# Resolving multisensory and attentional influences across cortical depth in sensory cortices

Remi Gau[1,2,3]*, Pierre-Louis Bazin[4,5], Robert Trampel[4], Robert Turner[4,6], Uta Noppeney[1,7]

[1]Computational Neuroscience and Cognitive Robotics Centre, University of Birmingham, Birmingham, United Kingdom; [2]Institute of Psychology, Université Catholique de Louvain, Louvain-la-Neuve, Belgium; [3]Institute of Neuroscience, Université Catholique de Louvain, Louvain-la-Neuve, Belgium; [4]Max Planck Institute for Human Cognitive and Brain Sciences, Leipzig, Germany; [5]Integrative Model-based Cognitive Neuroscience research unit, University of Amsterdam, Amsterdam, Netherlands; [6]Sir Peter Mansfield Imaging Centre, University of Nottingham, Nottingham, United Kingdom; [7]Donders Institute for Brain, Cognition and Behaviour, Radboud University, Nijmegen, Netherlands

**Abstract** In our environment, our senses are bombarded with a myriad of signals, only a subset of which is relevant for our goals. Using sub-millimeter-resolution fMRI at 7T, we resolved BOLD-response and activation patterns across cortical depth in early sensory cortices to auditory, visual and audiovisual stimuli under auditory or visual attention. In visual cortices, auditory stimulation induced widespread inhibition irrespective of attention, whereas auditory relative to visual attention suppressed mainly central visual field representations. In auditory cortices, visual stimulation suppressed activations, but amplified responses to concurrent auditory stimuli, in a patchy topography. Critically, multisensory interactions in auditory cortices were stronger in deeper laminae, while attentional influences were greatest at the surface. These distinct depth-dependent profiles suggest that multisensory and attentional mechanisms regulate sensory processing via partly distinct circuitries. Our findings are crucial for understanding how the brain regulates information flow across senses to interact with our complex multisensory world.

*For correspondence:
remi_gau@hotmail.com

Competing interests: The authors declare that no competing interests exist.

## Introduction

In our natural environment, our senses are exposed to a constant influx of sensory signals that arise from many different sources. How the brain flexibly regulates information flow across the senses to enable effective interactions with the world remains unclear.

Mounting evidence from neuroimaging (*Beauchamp et al., 2004*; *Noesselt et al., 2007*; *Rohe and Noppeney, 2016*), neurophysiology (*Atilgan et al., 2018*; *Kayser et al., 2010*; *Kayser et al., 2008*; *Lakatos et al., 2007*) and neuroanatomy (*Falchier et al., 2002*; *Rockland and Ojima, 2003*) suggests that interactions across the senses are pervasive in neocortex, arising even in primary cortices (*Driver and Noesselt, 2008*; *Ghazanfar and Schroeder, 2006*; *Liang et al., 2013*; *Schroeder and Foxe, 2002*). Visual stimuli can directly drive as well as modulate responses in cortices that are dedicated to other sensory modalities. Most prominently, functional magnetic resonance imaging (fMRI) in humans has shown that visual stimuli can induce crossmodal deactivations in primary and secondary auditory cortices (*Laurienti et al., 2002*; *Leitão et al., 2013*; *Mozolic et al., 2008*), yet enhance the response to a concurrent auditory stimulus (*Werner and Noppeney, 2011*; *Werner and Noppeney, 2010*). Further, neurophysiological research has suggested that

multisensory influences emerge early in sensory cortices and are to some extent preserved in anaesthetized animals (*Butler et al., 2012*; *Ibrahim et al., 2016*; *Iurilli et al., 2012*; *Kayser et al., 2007*; *Mercier et al., 2013*). Yet, the ability to extrapolate from neurophysiological findings in animals to human fMRI studies is limited by the nature of the BOLD response, which pools neural activity over time and across a vast number of neurons (*Logothetis, 2008*).

Information flow is regulated not only by multisensory but also by attentional mechanisms that are guided by our current goals (*Fairhall and Macaluso, 2009*; *Talsma et al., 2010*). Critically, multisensory and attentional mechanisms are closely intertwined. Both enhance perceptual sensitivity (*Leo et al., 2011*) and precision of sensory representations (*Ernst and Bülthoff, 2004*; *Fetsch et al., 2012*; *Meijer et al., 2019*; *Rohe and Noppeney, 2015*). Most importantly, the co-occurrence of two congruent sensory stimuli boosts the salience of an event (*Lewis and Noppeney, 2010*; *Van der Burg et al., 2008*), which may thereby attract greater attention. Conversely, a stimulus presented in one sensory modality alone may withdraw attentional resources from other sensory modalities. Behavioural and functional imaging studies have shown that shifting attention endogenously to one sensory modality reduces processing and activations in the unattended sensory systems (*Ciaramitaro et al., 2007*; *Johnson and Zatorre, 2005*; *Mozolic et al., 2008*; *Rohe and Noppeney, 2018*; *Rohe and Noppeney, 2015*; *Shomstein and Yantis, 2004*). As a consequence, attentional mechanisms may contribute to competitive and cooperative interactions across the senses, for instance by amplifying responses for congruent audiovisual stimuli, and generating crossmodal deactivations for unisensory stimuli. Inter-sensory attention can also profoundly modulate multisensory interactions (*Talsma et al., 2007*). Most prominently, the influence of visual stimuli on auditory cortices was shown to be enhanced when attention was focused on the visual sense (*Lakatos et al., 2009*).

While previous neurophysiological studies have revealed influences of modality-specific attention predominantly in superficial laminae in non-human primates (*Lakatos et al., 2009*), visual influences on auditory cortices have recently been shown to be most prominent in deep layer 6 of auditory cortex in rodents (*Morrill and Hasenstaub, 2018*). In other words, combined neurophysiological evidence from primates and rodents suggests a double disassociation of attentional and multisensory influences.

To investigate whether this double dissociation can be found in human neocortex, we exploited recent advances in submillimeter-resolution fMRI at 7T that allow the characterization of depth-dependent activation profiles (*De Martino et al., 2015a*; *Duong et al., 2003*; *Harel et al., 2006*; *Kok et al., 2016*; *Koopmans et al., 2010*; *Muckli et al., 2015*; *Polimeni et al., 2010*; *Trampel et al., 2012*). While gradient-echo echo-planar-imaging (GE-EPI) BOLD fMRI is not yet able to attribute activations unequivocally to specific cortical layers (*Duong et al., 2003*; *Goense et al., 2012*; *Harel et al., 2006*; *Huber et al., 2017*; *Markuerkiaga et al., 2016*; *Trampel et al., 2019*), the observation in the same cortical territories of distinct laminar activation profiles induced by multisensory and attentional influences would strongly imply distinct neural mechanisms.

The current study investigated the processing of auditory, visual or audiovisual looming stimuli under auditory and visual attention in the human brain. Combining submillimeter-resolution fMRI at 7T with laminar and multivariate pattern analyses, we show distinct depth-dependent activation profiles and/or patterns for multisensory and attentional influences in early auditory and visual cortices. These results suggest that multisensory and attentional mechanisms regulate sensory processing in early sensory cortices via partly distinct neural circuitries.

## Results

In this fMRI study, participants were presented with blocks of auditory (A), visual (V) and audiovisual (AV) looming stimuli interleaved with fixation (*Figure 1A*). We used looming motion as a biologically relevant and highly salient stimulus that reliably evokes crossmodal influences in sensory cortices in human neuroimaging and animal neurophysiology (*Cappe et al., 2012*; *Maier et al., 2008*; *Tyll et al., 2013*). Modality-specific attention was manipulated by requiring participants to detect and respond selectively to weak auditory or visual targets, which were adjusted prior to the main study to threshold performance in sound amplitude or visual size for each participant. The targets were interspersed throughout all auditory, visual and audiovisual looming blocks (e.g. visual targets were presented during both visual and auditory looming blocks; see *Figure 1B*).

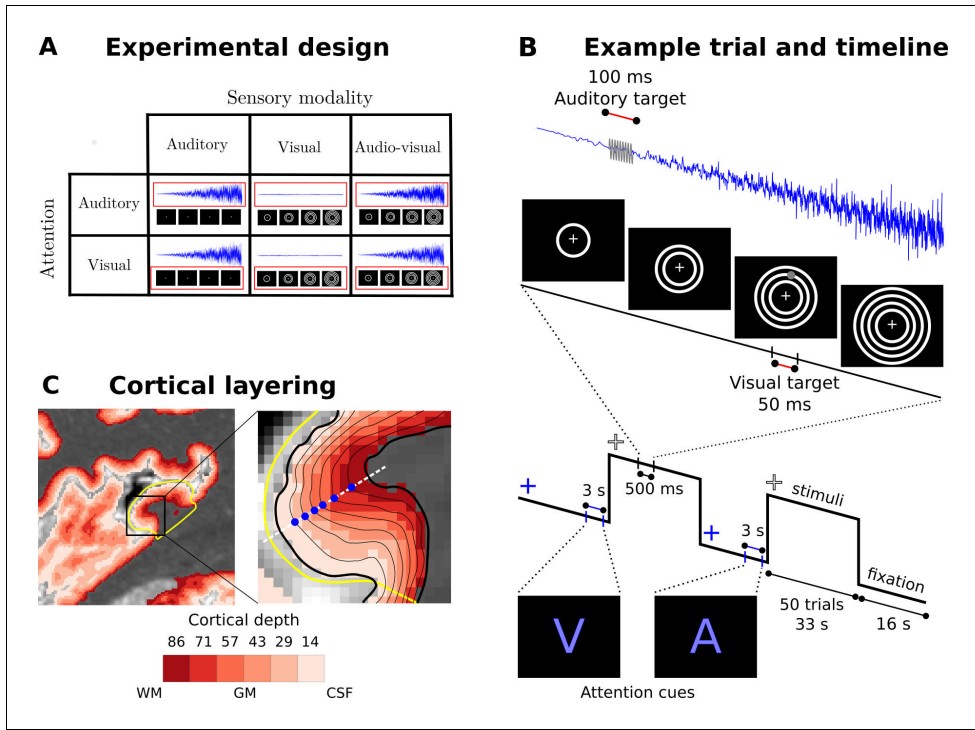

**Figure 1.** Experimental design, timeline and cortical layering. (**A**) Experimental design: Participants were presented with auditory, visual and audiovisual looming stimuli under auditory and visual attention. (**B**) Example trial and timeline: Participants were presented with brief auditory, visual or audiovisual looming stimuli in 33 s blocks interleaved with 16 s fixation. At the beginning of each block, a cue indicated whether the auditory or visual modality needed to be attended. Brief visual and auditory targets (grey) were interspersed in the looming activation blocks. Participants were instructed to respond to the targets in the attended and ignore the targets in the unattended sensory modality. (**C**) Cortical layering: Left: A parasagittal section of a high resolution T1 map is shown with a colour coded laminar label for each voxel (voxel size: $(0.4 \text{ mm})^3$). The primary auditory cortex is circled in yellow. Right: The cortical sheet is defined by the pial and white matter surfaces (thick black solid lines). Six additional surfaces (thin black solid lines) were determined at different cortical depths. Data were mapped onto those surfaces by sampling (blue dots) radially along the normal (white dashed line) to the mid-cortical depth surface (not shown here). WM: white matter, GM: grey matter, CSF: cerebrospinal fluid.

The online version of this article includes the following figure supplement(s) for figure 1:

**Figure supplement 1.** Behavioural results.
**Figure supplement 2.** Segmentation and coregistration.

For the behavioural analysis, the percentages of visual and auditory targets that gained a response (see *Figure 1—figure supplement 1* and *Supplementary file 1* for '% target responses') were entered into a 2 (target modality: auditory vs. visual) X 2 (attended modality: auditory vs. visual) X 3 (stimulus block modality: auditory, visual, audio-visual) repeated measures ANOVA. We observed significant main effects of attended modality ($F_{(1, 10)}$=291.263, p<0.001, $\eta p^2$ = 0.967), target modality ($F_{(1, 10)}$=13.289, p=0.004, $\eta p^2$ = 0.571), and stimulus modality ($F_{(1.986, 19.864)}$ =5.304, p=0.014, $\eta p^2$ = 0.347), together with a significant interaction between target modality and attended modality ($F_{(1, 10)}$=14.786, p=0.003, $\eta p^2$ = 0.597). This interaction confirmed that participants had successfully maintained auditory or visual attention as instructed.

Further, we observed significant interactions between target modality and stimulus block modality ($F_{(1.813, 18.126)}$=8.149, p=0.004, $\eta p^2$ = 0.449) and between attended modality and stimulus block modality ($F_{(1.436, 14.360)}$=24.034, p<0.001, $\eta p^2$ = 0.706). These interactions resulted from greater hit rates for auditory targets given auditory attention than for visual targets given visual attention during both auditory ($t_{(10)}$=2.845, p=0.017, Hedges $g_{av}$ = 0.804) and visual blocks ($t_{(10)}$ =4.432, p=0.001, Hedges $g_{av}$ = 2.037), but not for audio-visual blocks ($t_{(10)}$=0.276, p=0.788, Hedges $g_{av}$ = 0.081) suggesting that observers' performance was not completely matched across all

conditions. Specifically, the presentation of auditory and visual stimuli in the audiovisual blocks interfered with the detection of auditory targets. For completeness, there was no significant three-way interaction (F(1.783, 17.826)=2.467, p=0.118, $\eta p^2$ = 0.198).

Using sub-millimeter resolution fMRI, we characterized the laminar profiles in auditory (primary auditory cortex, A1; planum temporale, PT) and visual (primary, V1; higher order, V2/3) regions for the following effects: 1. sensory deactivations in unisensory contexts for non-preferred stimuli (i.e. crossmodal, e.g. [V-Fix] in auditory cortices), 2. crossmodal modulation (e.g. [AV-A] in auditory cortices, [AV-V] in visual cortices) in audiovisual context and 3. direct and modulatory effects of modality-specific attention.

Briefly, we used the following methodological approach (see Material and methods): in each ROI and participant, we estimated the regional BOLD response (i.e. 'B parameter estimate') (e.g. *Figure 2A* row 1, left) and the multivariate pattern decoding accuracy for each of the six laminae (e.g. *Figure 3A* row 1, right). We then characterized the laminar profiles of BOLD response and decoding accuracy in terms of a constant and a linear shape parameter (i.e. 'S parameter estimates') and show their across-subjects' mean and distribution in violin plots (e.g. *Figure 2A* row 2). Finally, we characterized the spatial topography of those S parameter estimates by projecting their group mean onto a normalized group cortical surface (e.g. *Figure 2B*).

All statistical results are presented in *Table 1*, *Table 2*, *Table 3* and *Supplementary files 3* and *4*. Additional descriptive statistics as well as effect sizes can be found here: https://osf.io/tbh37/.

## Auditory cortices

Auditory stimuli evoked a positive BOLD response in primary auditory cortex and especially in anterior portions of the planum temporale (*Figure 2—figure supplements 1B* and *3* right). As expected from the typical physiological point spread function of the GE-EPI BOLD signal, the positive BOLD signal increased roughly linearly towards the cortical surface (*Markuerkiaga et al., 2016*) (*Figure 2—figure supplement 1A* and *Supplementary file 3*).

Deactivations induced by crossmodal visual stimuli (i.e. a negative BOLD response for [V-Fix]) were observed in both A1 and PT with a constant response profile and based on visual inspection even a trend towards stronger deactivations in deeper laminae (*Figure 2A* left, *Table 1*). These visually induced deactivations were generally observed for both auditory and visual attention conditions, with no significant difference between them ([Att$_A$-Att$_V$] for visual stimuli in A1 and PT: F (2,40) = 0.644, p=0.530).

We did not observe a significant crossmodulatory effect of visual stimuli on the BOLD-response in A1 and PT in the context of concurrent auditory stimuli (*Figure 3A* left, *Table 2A*). However, pattern classifiers succeeded in discriminating between patterns elicited by [AV vs. A] conditions across all laminae in both PT and A1 (*Figure 3A* right, *Table 2B*). Thus, even when the mean BOLD response across vertices averaged across A1 and PT did not differ significantly for AV and A stimuli at a particular depth, the visual stimulus changed the activation pattern elicited by a concurrent auditory stimulus. This provides evidence that sub-regions with crossmodal enhancement and suppression co-exist in A1 and PT. The visual induced changes in activation patterns were again not significantly affected by modality-specific attention (for the classification [AV-V]$_{att\ A}$ VS [AV-V]$_{att\ V}$, F(2,40) = 0.185, p=0.832).

Taken together these results suggest that salient visual looming stimuli affected auditory cortices irrespective of the direction of modality-specific attention in both unisensory and audiovisual contexts. Further, the surface projections of the shape parameters of the BOLD response profile at the group level revealed a patchy topography both for visually-induced deactivations in a unisensory context (*Figure 2B* left) and for visual-induced modulations of the auditory response (*Figure 3B* left i and ii).

We next explored whether visual stimuli influenced activation patterns in auditory cortices in a similar patchy topography during unisensory visual and audiovisual contexts. For this, we defined a general linear model for each subject that used the 'constant' or 'linear' shape parameters from the visual deactivations (i.e. [V-Fix]) for each vertex to predict the 'constant', and respectively 'linear', shape parameters for the visually induced crossmodal modulation (i.e. [AV-A]) over vertices (again in a cross-validated fashion). We visualized the results of this regression in the form of raster plots (*Figure 3B* right i and ii). These raster plots show the laminar BOLD response profile for the difference [AV-A] across vertices, sorted according to their BOLD response profile for [V-fix].

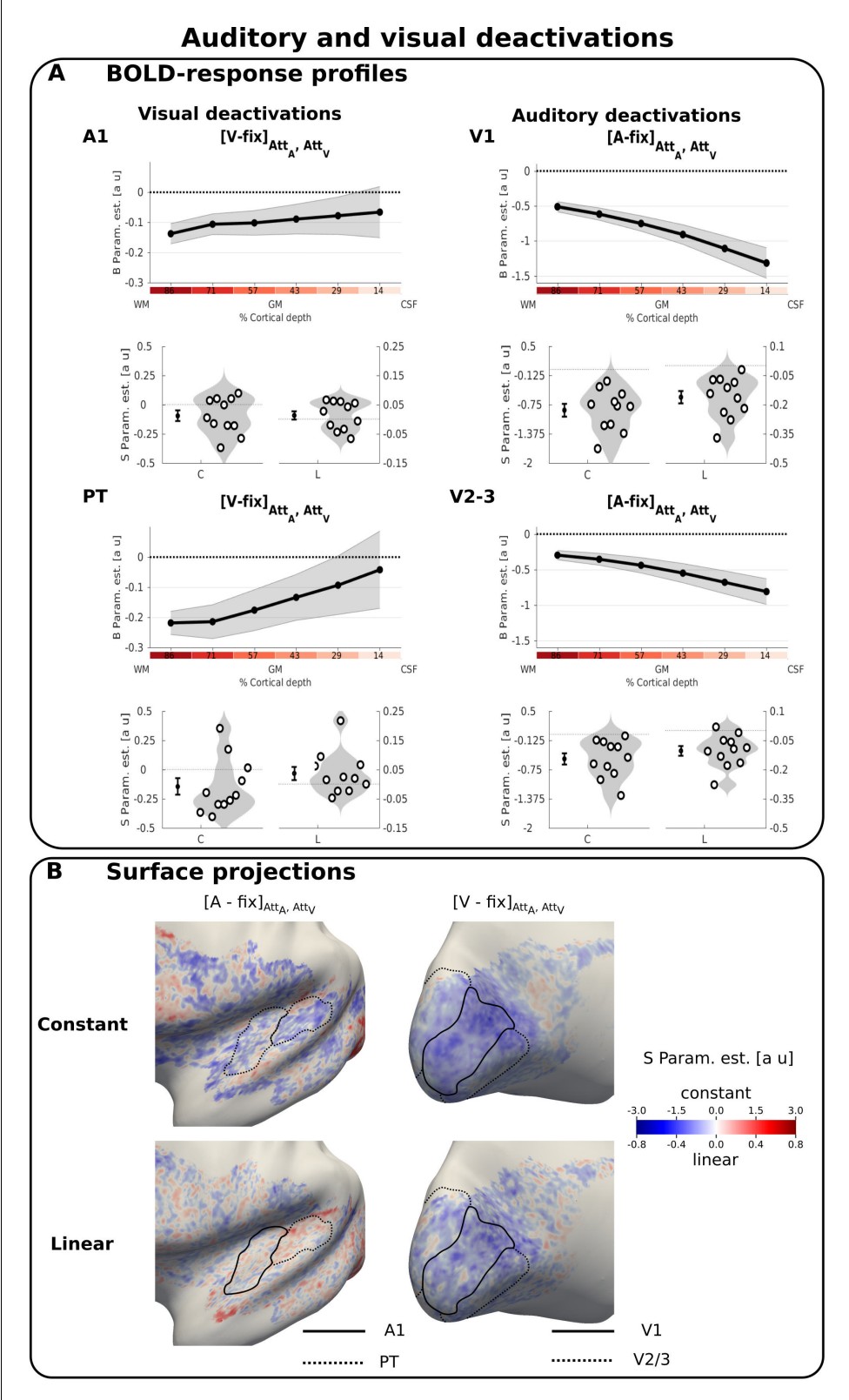

**Figure 2.** Auditory and visual deactivations. (**A**) BOLD response profiles: Rows 1 and 3: The BOLD response (i.e. B parameters, across subjects' mean ± SEM) for visual and auditory looming stimuli averaged over auditory and visual attention in A1, PT, V1, V2/3 is shown as a function of percentage cortical depth. WM: white matter; GM: grey matter; CSF: cerebrospinal fluid. Percentage cortical depth is indicated by the small numbers and colour coded in red. Rows 2 and 4: Across subjects' mean (± SEM) and violin plot of the participants' shape parameter estimates that characterize

*Figure 2 continued on next page*

*Figure 2 continued*

the mean (C: constant) and linear increase (L: linear) of the laminar BOLD response profile. n = 11. (**B**) Surface projections: Across subject' mean of the 'constant' (row i) and 'linear' increase (row ii) shape parameter estimates of the laminar BOLD response profile for auditory and visual looming stimuli (averaged over auditory and visual attention) are projected on an inflated group mean surface to show auditory and visual regions of the left hemisphere. A1 and V1 are delineated by black solid lines, PT and V2/3 by dashed lines. For visualization purposes only: i. surface projections were smoothed (FWHM = 1.5 mm); ii. values are also presented for vertices for which data were not available from all subjects and which were therefore not included in our formal statistical analysis. Grey areas denote vertices with no available data for any subject.

The online version of this article includes the following figure supplement(s) for figure 2:

**Figure supplement 1.** Auditory and visual activations.
**Figure supplement 2.** Auditory and visual responses in visual areas.
**Figure supplement 3.** Auditory and visual responses in auditory areas.

In A1 and PT, the shape profile of a vertex for visual deactivations significantly predicted its profile for cross-modal modulation suggesting similar patchy topographies for visual deactivations and crossmodal modulation. In PT, the constant parameter for [V-Fix] predicted the constant for [AV-A]: the less a vertex is deactivated in PT by visual stimuli in a unisensory context, the more it shows a visually induced enhancement of the response to a concurrent auditory stimulus (constant: t(10) =3.460, p=0.006, linear: t(10)=1.853, p=0.094, see *Figure 3Bi* right). More interestingly, in A1 the linear shape parameter for [V-Fix] predicted the linear shape parameter of [AV-A]: vertices with less deactivations in deeper relative to superficial laminae in unisensory visual context showed a robust crossmodal enhancement that was most pronounced in deeper laminae (linear: t(10)=3.121, p=0.011, constant: t(10)=0.021, p=0.983, *Figure 3Bii* right). These results strongly suggest that visual stimuli generate a BOLD response in primary auditory cortex with a patchy topography similar for unisensory and audiovisual contexts.

Because these commonalities in topography across two contrasts could in principle arise from spurious factors (such as registration errors, curvature-dependent segmentation errors, heterogeneous occurrence of principal veins, curvature dependent occurrence of veins, orientation dependence to B0, signal leakage of kissing gyri) that could affect BOLD-response magnitude similarly in different contrasts, we repeated the statistical analysis and raster plots for the constant shape parameters of the [V-Fix] and [A-Fix] contrasts in auditory areas. However, as shown in the raster plots in *Figure 3—figure supplement 2* this control analysis did not reveal any significant relationship between the two contrasts. The absence of a significant effect in this control analysis thus suggests that the similarity in topography between visually induced deactivations and cross-modal modulations reflects similarities in neural organization rather than non-specific effects.

In contrast to these multisensory influences, attentional modulation was greatest at the cortical surface in both A1 and PT for the regional BOLD response (i.e. significant positive linear effect in A1 and PT, *Figure 4Ai* left, *Table 3A*). Averaged across A1 and PT, pattern classifiers also discriminated between the auditory and visual attention conditions (*Figure 4Ai* right, *Table 3B*). The decoding accuracy profiles were characterized by a significant constant term and a non-significant trend for the linear term. Surprisingly, even though the mean BOLD-response profiles revealed a profound effect of attention, the discrimination between auditory and visual attention conditions was rather limited. Discriminating between activation patterns may be limited, if modality-specific attention predominantly amplifies and scales the BOLD response for A, V or AV stimuli whilst preserving the activation patterns, whereas the crossmodal influences impacts the activation patterns.

To investigate this explanation further, we quantified and compared the similarity in A1 between activation patterns of auditory vs. visual attention conditions (averaging over i. $A\_Att_V$ vs. $A\_Att_A$; and ii. $AV\_Att_V$ vs. $AV\_Att_A$) and of auditory vs. audiovisual conditions (averaging over iii. $A\_Att_A$ vs. $AV\_Att_A$; and iv. $A\_Att_V$ vs. $AV\_Att_V$) using the Spearman correlation coefficient computed over vertices. The average similarity between auditory and visual attention conditions was indeed greater than the average similarity between A and AV conditions (two-sided exact sign permutation test on Fisher transformed correlation coefficients, p=0.008). These results suggest that auditory attention mainly scales the BOLD response in A1 whilst preserving the activation pattern, whereas visual stimuli alter the activation pattern in A1.

In summary, we observed distinct laminar BOLD response profiles and activation patterns for multisensory and attentional influences in auditory cortices. Visual stimuli induced deactivations with a

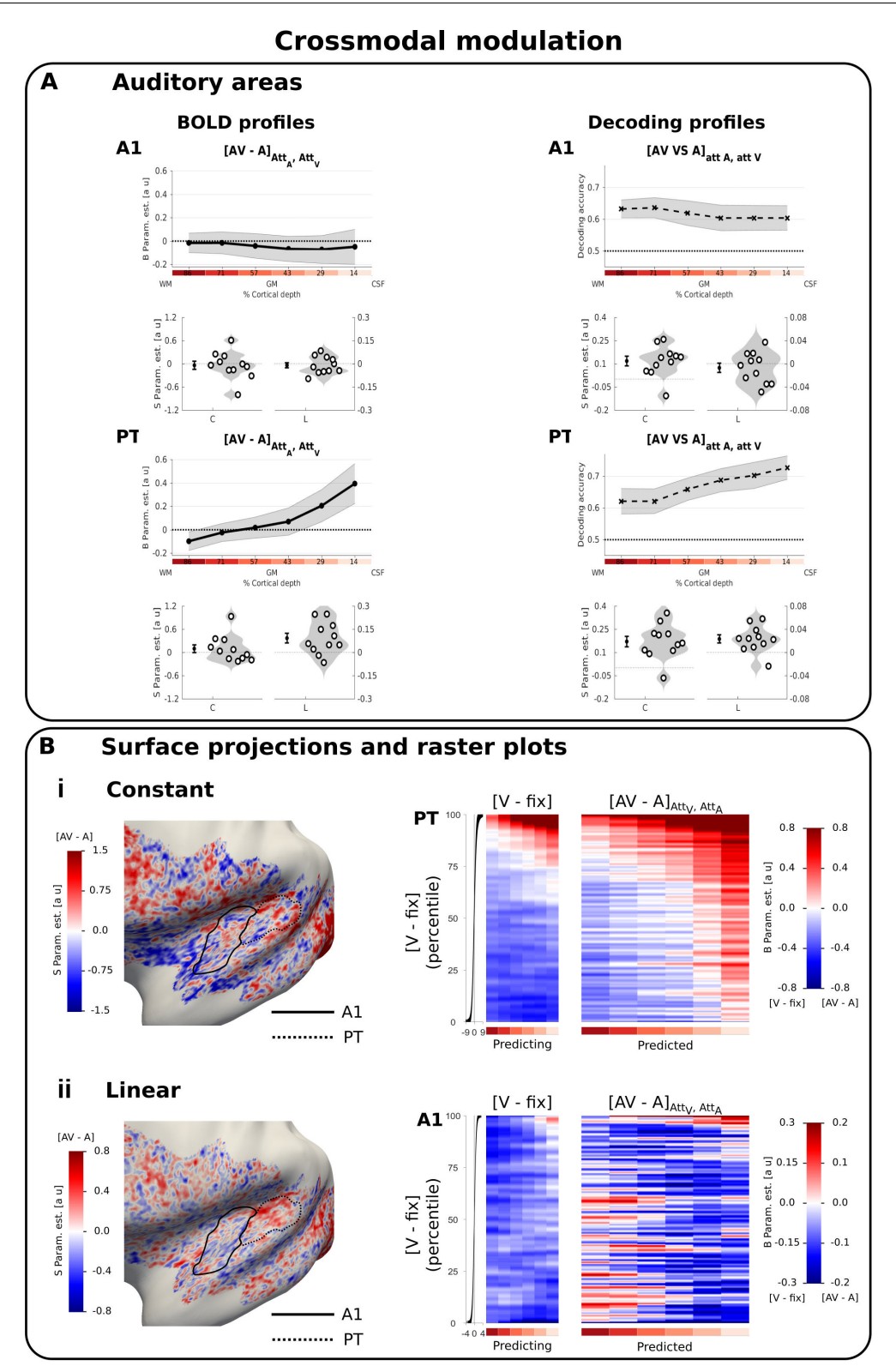

**Figure 3.** Cross-modal modulation in auditory areas. (**A**) Laminar profiles: Rows 1 and 3: The BOLD response (solid line; column 1 and 3) and decoding accuracy (dashed line; columns 2 and 4) (across subjects' mean ± SEM) for [AV-A] in A1 and PT is shown as a function of percentage cortical depth pooled (i.e. averaged) over auditory and visual attention. WM: white matter, GM: grey matter, CSF: cerebrospinal fluid. Percentage cortical depth is indicated by the small numbers and colour coded in red. Rows 2 and 4: Across subjects' mean (± SEM) and violin plot of participants' shape parameter
*Figure 3 continued on next page*

*Figure 3 continued*

estimates that characterize the mean (C: constant) and linear increase (L: linear) of the laminar BOLD response and decoding accuracy profiles. n = 11. (B) Surface projections and raster plots: **Left**: Across subject' mean of the 'constant' (row i) and 'linear increase' (row ii) shape parameter estimates of the laminar BOLD response profile for [AV-A] (averaged over auditory and visual attention) are projected on an inflated group mean surface to show auditory regions of the left hemisphere. A1 and PT are delineated by black solid and dashed lines. For visualization purposes only: i. surface projections were smoothed (FWHM = 1.5 mm); ii. values are also presented for vertices for which data were not available from all subjects and which were therefore not included in our formal statistical analysis. Grey areas denote vertices with no available data for any subject. Right: Row i. PT: The raster plot illustrates the statistical relationship between the 'constant' shape parameters for the visual evoked response [V-Fix]$_{AttA, AttV}$ and the crossmodal modulation [AV-A]$_{AttA, AttV}$ in PT. Each raster plot shows the laminar profiles (colour coded along abscissa) of the vertices for the 'predicting contrast' [V-Fix] and of the 'predicted contrast' [AV-A]. The laminar profiles of the vertices were sorted along the ordinate according to the value of the 'constant' shape parameter for [V-Fix]. The raster plot shows that the laminar profile of a vertex for [V-Fix] can predict its laminar profile for [AV-A]: PT vertices with less deactivations across laminae for [V-Fix] are associated with greater crossmodal enhancement [AV-A]. Row ii. The raster plot illustrates the statistical relationship between the 'linear slope' shape parameters for the visual evoked response [V-Fix]$_{AttA, AttV}$ and the crossmodal modulation [AV-A] in A1. Each raster plot shows the laminar profiles (colour coded along abscissa) of the vertices for the 'predicting contrast' [V-Fix] and of the 'predicted contrast' [AV-A]. The laminar profiles of the vertices are sorted along the ordinate according to the value of the 'linear' shape parameter for [V-Fix]. A1 vertices with less deactivations in deeper laminae for [V-Fix] are associated with greater crossmodal enhancement [AV-A] in deeper laminae. To enable averaging the raster plots across participants, the vertices were binned after sorting (number of bins for A1: 10100, number of bins for PT: 6800). For visualization purposes all raster plots were smoothed along the vertical axis (FWHM = 1% of the number of data bins). The subplot (i.e. black line) to the left of the raster plots shows the across subjects' mean value (+ /- STD) of the shape parameters (i.e. i. constant, ii, linear) of the sorting contrast. n = 11.

The online version of this article includes the following figure supplement(s) for figure 3:

**Figure supplement 1.** Cross-modal modulation in visual areas.

**Figure supplement 2.** Raster plots for auditory induced activations and visual induced deactivations in auditory areas.

constant profile both in A1 and PT. Likewise, they influenced the activation pattern in auditory cortices similarly across all laminae (i.e. constant effect for decoding accuracy) in audiovisual context leading to successful pattern decoding accuracy for [AV vs. A] even in deeper laminae. In fact, in A1 and PT visual inputs altered the activation pattern with a similar patchy topography when presented alone (i.e. visual-induced deactivation) or together with an auditory stimulus (i.e. crossmodal modulation).

Modality-specific attention amplified the mean BOLD-signal within A1 and PT mainly at the cortical surface (i.e. significant linear parameter). Likewise the decoding accuracy was better than chance with a non-significant increase toward the surface.

**Table 1.** Auditory and visual deactivations.

| | | Linear or constant | | | Constant | | Linear | |
|---|---|---|---|---|---|---|---|---|
| [V-fix]Att_A, Att_V | Mean(A1, PT) | F(2,40) = 9.280 | p<0.001 | | t(10)=−2.460 | p=0.017* | F(1,20) = 2.083 | p=0.164 |
| | | | | A1 | t(10)=−2.077 | p=0.032* | | |
| | | | | PT | t(10)=−2.042 | p=0.034* | | |
| [A-fix]Att_A, Att_V | mean(V1, V23) | F(2,40) = 58.615 | p<0.001 | | t(10)=−5.547 | p<0.001* | F(1,20) = 22.433 | p<0.001 |
| | | | | V1 | t(10)=−6.538 | p<0.001* | t(10)=−5.080 | p<0.001 |
| | | | | V2-3 | t(10)=−4.305 | p<0.001* | t(10)=−4.142 | p=0.002 |

*indicates p-values based on a one-sided t-test based on a priori hypotheses. p-values<0.05 are indicated in bold. n = 11

Using 2 (shape parameter: constant, linear) x 2 (ROI: primary, non-primary) linear mixed effects models, we performed the following statistical comparisons in a 'step down procedure':

1. Two-dimensional F-test assessing whether the constant or linear parameter (e.g. each averaged across ROIs in auditory resp. visual cortices), was significantly different from zero (dark grey),

2. If this two-dimensional F-test was significant, we computed one dimensional F-tests separately for the constant and the linear parameters (again averaged across auditory resp. visual ROIs) (light grey),

3. If the one dimensional F-test was significant, we computed follow-up t-tests separately for each of the two ROIs (white).

**Table 2.** Effects of the cross-modal modulation on the laminar BOLD response and decoding accuracy profiles in auditory areas.

**A) BOLD profile**

|  |  | linear or constant |  |  | constant |  | linear |  |
| --- | --- | --- | --- | --- | --- | --- | --- | --- |
| [AV - A]Att_A, Att_V | mean(A1, PT) | F(2,40) = 0.196 | p=0.823 |  |  |  |  |  |

**B) Decoding profile**

|  |  | linear or constant |  |  | constant |  | linear |  |
| --- | --- | --- | --- | --- | --- | --- | --- | --- |
| [AV VS A]att A, att V | mean(A1, PT) | F(2,40) = 34.946 | p<0.001 |  | F(1,20) = 21.966 | p<0.001 | F(1,20) = 1.850 | p=0.189 |
|  |  |  |  | A1 | t(10)=3.867 | p=0.003 |  |  |
|  |  |  |  | PT | t(10)=4.992 | p<0.001 |  |  |

Using 2 (shape parameter: constant, linear) x 2 (ROI: primary, non-primary) linear mixed effects models, we performed the following statistical comparisons in a 'step down procedure':

1. Two-dimensional F-test assessing whether the constant or linear parameter (e.g. each averaged across ROIs in auditory resp. visual cortices), was significantly different from zero (dark grey),

2. If this two-dimensional F-test was significant, we computed one dimensional F-tests separately for the constant and the linear parameters (again averaged across auditory resp. visual ROIs) (light grey),

3. If the one dimensional F-test was significant, we computed follow-up t-tests separately for each of the two ROIs (white).

## Visual cortices

In V1 and V2/3 visual stimuli induced activations that increased toward the cortical surface as expected for GE-EPI BOLD (*Markuerkiaga et al., 2016*) (*Figure 2—figure supplements 1* and *2* left and *Supplementary file 1*). Contrary to previous research (*Chen et al., 2013*; *Koopmans et al., 2010*), we did not observe a more selective increase in activations for layer 4. Potentially, this selective increase in BOLD-response may have been smoothed across multiple layers, because the

**Table 3.** Effects of the attentional modulation (irrespective of stimulus type) on the laminar BOLD response and decoding accuracy profiles.

**A) BOLD profile**

|  |  | linear or constant |  |  | constant |  | linear |  |
| --- | --- | --- | --- | --- | --- | --- | --- | --- |
| [Att_V - Att_A]A, V, AV | mean(A1, PT) | F(2,40) = 12.602 | p<0.001 |  | F(1,20) = 9.249 | p=0.006 | F(1,20) = 12.163 | p=0.002 |
|  |  |  |  | A1 | t(10)=1.882 | p=0.089 | t(10)=3.123 | p=0.011 |
|  |  |  |  | PT | t(10)=4.523 | p=0.001 | t(10)=3.361 | p=0.007 |
| [Att_V - Att_A]A, V, AV | mean(V1, V23) | F(2,40) = 0.669 | p=0.518 |  |  |  |  |  |

**B) Decoding profile**

|  |  | linear or constant |  |  | constant |  | linear |  |
| --- | --- | --- | --- | --- | --- | --- | --- | --- |
| [Att_A VS Att_V]A, V, AV | mean(A1, PT) | F(2,40) = 4.687 | p=0.015 |  | F(1,20) = 4.882 | p=0.039 | F(1,20) = 4.028 | p=0.058 |
|  |  |  |  | A1 | t(10)=1.260 | p=0.236 |  |  |
|  |  |  |  | PT | t(10)=2.031 | p=0.070 |  |  |
| [Att_A VS Att_V]A, V, AV | mean(V1, V23) | F(2,40) = 20.026 | p<0.001 |  | F(1,20) = 13.564 | p=0.001 | F(1,20) = 9.951 | p=0.005 |
|  |  |  |  | V1 | t(10)=2.472 | p=0.033 | t(10)=1.359 | p=0.204 |
|  |  |  |  | V2-3 | t(10)=4.298 | p=0.002 | t(10)=3.089 | p=0.011 |

Using 2 (shape parameter: constant, linear) x 2 (ROI: primary, non-primary) linear mixed effects models, we performed the following statistical comparisons in a 'step down procedure':

1. Two-dimensional F-test assessing whether the constant or linear parameter (e.g. each averaged across ROIs in auditory resp. visual cortices), was significantly different from zero (dark grey),

2. If this two-dimensional F-test was significant, we computed one dimensional F-tests separately for the constant and the linear parameters (again averaged across auditory resp. visual ROIs) (light grey),

3. If the one dimensional F-test was significant, we computed follow-up t-tests separately for each of the two ROIs (white).

p-values<0.05 are indicated in bold. n = 11.

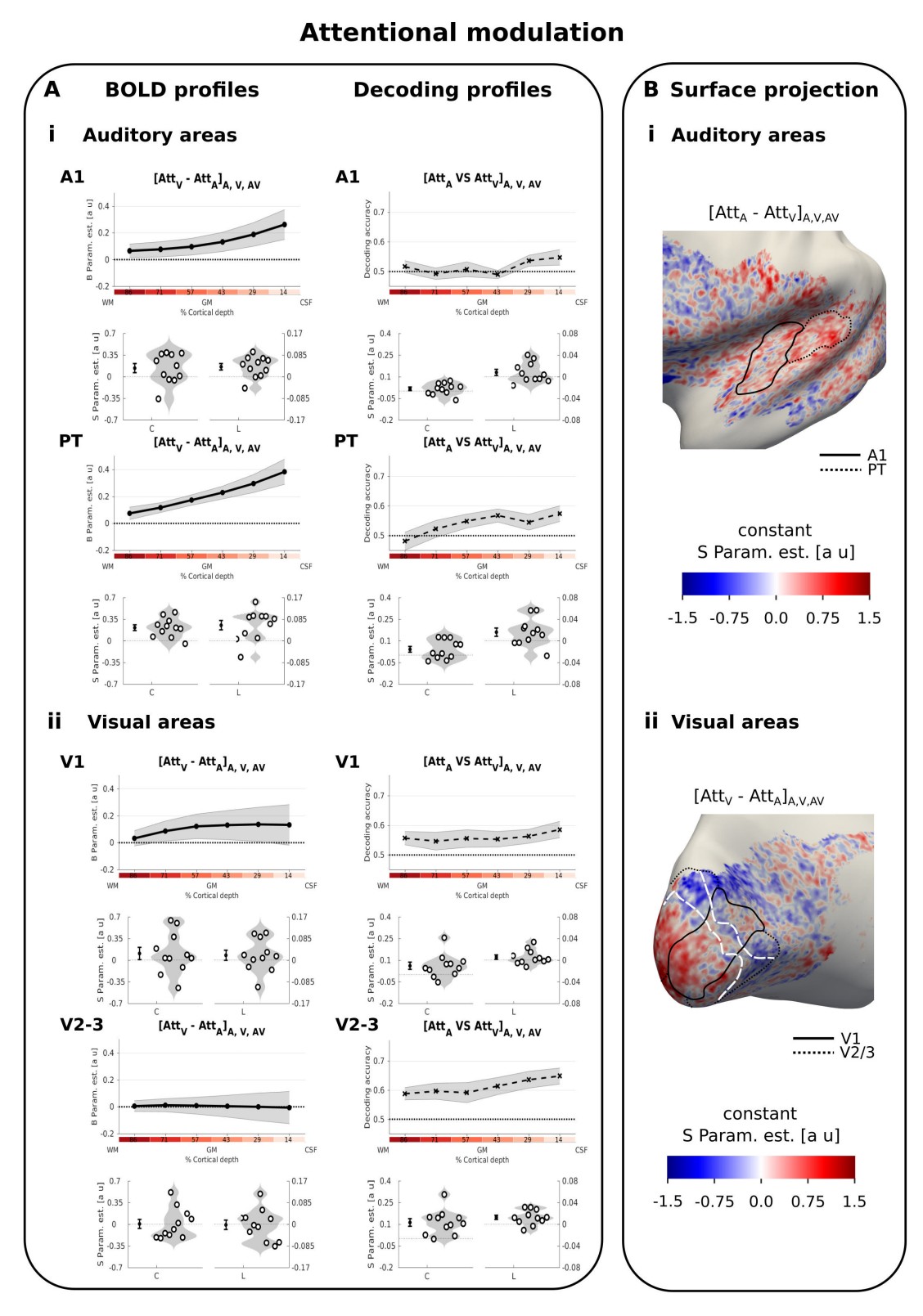

**Figure 4.** Attentional modulation. (**A**) Laminar profiles: Rows 1, 3, 5, 7: The BOLD response (solid line; columns 1 and 3) and decoding accuracy (dashed line; columns 2 and 4) (across subjects' mean ± SEM) for attentional modulation (i: top - [Att_A-Att_V] in A1 and PT; ii: bottom - [Att_V-Att_A] in V1 and V2/3) is shown as a function of percentage cortical depth pooled (i.e. averaged) over auditory, visual and audiovisual looming stimuli. WM: white matter, GM: grey matter, CSF: cerebrospinal fluid. Percentage cortical depth is indicated by the small numbers and colour coded in red. Rows 2, 4, 6, 8: Across

*Figure 4 continued on next page*

*Figure 4 continued*

subjects' mean (± SEM) and violin plot of the participants' shape parameter estimates that characterize the mean (C: constant) and linear increase (L: linear) of the laminar BOLD response and decoding accuracy profiles. n = 11. B. Surface projections: Across subject' mean of the 'constant' shape parameter estimates for attentional modulation [Att$_A$-Att$_V$]$_{A,V, AV}$ in A1 and PT (i: top) and in V1 and V2/3 (ii: bottom) are projected on an inflated group mean surface of the left hemisphere. A1 and V1 are delineated by black solid lines. PT and V2/3 are delineated by dashed lines. For visualization purposes only: i. surface projections were smoothed (FWHM = 1.5 mm); ii. values are also presented for vertices for which data were not available from all subjects and which were therefore not included in our formal statistical analysis; iii borders between visual-induced activations and deactivations (white dashed lines on the right) are reported here from *Figure 2B*. Grey areas denote vertices with no available data for any subject.

relative cortical depth of layer four varies between foveal and lateral projections in V1 and our region of interest is larger than those used in previous work (for similar findings see figure 6B in *Polimeni et al., 2010* for fMRI and figure 6E in *Waehnert et al. (2016)* for structural anatomy). Consistent with previous research (*Shmuel et al., 2002*; *Silver et al., 2008*; *Tootell et al., 1998*), the centrally presented visual looming stimuli activated predominantly areas representing the centre of the visual field with response suppressions (i.e. negative BOLD) in adjacent areas (see dashed white lines in *Figure 2—figure supplement 1B* right). By contrast, auditory deactivations were not confined to the central field that was activated by visual stimuli. Instead, they were particularly pronounced in the rostral part of the calcarine sulcus representing more eccentric positions of the visual field (*Figure 2B* right and *Figure 2—figure supplement 2* right). Furthermore, auditory looming stimuli induced deactivations that increased in absolute magnitude toward the cortical surface (significant constant and linear effect, *Table 1*). Again as in auditory areas, these auditory deactivations were not significantly modulated by modality-specific attention ([Att$_V$-Att$_A$] for auditory stimuli in V1 and V2/3: F(2,40) = 1.651, p=0.205).

Despite the pronounced deactivations elicited by unisensory auditory stimuli, we did not observe any significant cross-modal modulation of the regional BOLD response in V1 and V2/3 (see *Figure 3—figure supplement 1* left and *Supplementary file 4A*). Likewise, the support vector classifiers were not able to discriminate between activation patterns for [AV vs. V] stimuli better than chance. In summary, the effect of auditory stimuli on visual cortices was abolished under concurrent visual stimulation both in terms of regional BOLD response and activation patterns (see *Figure 3—figure supplement 1* right and *Supplementary file 4B*).

With respect to attentional modulation, we observed no significant differences in mean BOLD responses in V1 or V2/3 for visual vs. auditory attention conditions (see *Table 3A*). As shown in the violin plots of *Figure 4Aii* left, only few participants showed a substantial attentional modulation effect when averaged across the entire regions of interest. Yet, the classifier was able to discriminate between patterns for visual and auditory attention conditions successfully across all cortical depth surfaces in V1 and V2/3 (i.e. significant constant term in V1 and V2/3 and linear term in V2/3, see *Table 3B*, *Figure 4Aii* right). The surface projection of shape parameters of the BOLD response profile at the group level explains this discrepancy between the attentional effects on regional BOLD response and activation patterns (*Figure 4Bii*). Consistent with mechanisms of contrast enhancement (*Bressler et al., 2013*; *Müller and Kleinschmidt, 2004*; *Smith et al., 2000*; *Tootell et al., 1998*), attending to vision relative to audition increased responses in regions that were activated by visual looming stimuli, yet suppressed the deactivations in the surrounding regions that were deactivated by visual stimuli thereby cancelling out global attentional effects for the regional BOLD response (*Figure 4Bii*, note the white lines indicate the border between visual activation and deactivation from *Figure 2—figure supplement 1B* right).

In summary, auditory looming induced widespread deactivations in visual cortices that were maximal at the cortical surface and extended into regions that represent more peripheral visual fields. By contrast, attention to vision predominantly increased BOLD response in central parts that were activated by visual stimuli and suppressed BOLD responses in the periphery that were deactivated by visual stimuli.

## Discussion

The current high-resolution 7T fMRI study revealed distinct depth-dependent BOLD response profiles and patterns for multisensory and attentional influences in early sensory cortices.

## Distinct laminar profiles and activation patterns for multisensory and attentional influences in auditory cortices

Visual looming suppressed activations in auditory cortices (see also [*Laurienti et al., 2002*; *Leitão et al., 2013*; *Mozolic et al., 2008*], but enhanced the response to concurrent auditory looming in posterior auditory cortices (*Werner and Noppeney, 2011*; *Werner and Noppeney, 2010*). In other words, intersensory competition for purely visual stimuli turned into cooperative interactions for audiovisual stimuli. While the visual-induced response suppression was constant across cortical depth in A1 and PT, the crossmodal BOLD-response enhancement was observed mainly in the caudal parts of PT, that is in parts of PT where visual influences have previously been shown (*Kayser et al., 2007*). At a finer spatial resolution, multivariate analyses revealed significant differences between audiovisual and auditory activation patterns across all laminae in both A1 and PT. As shown in the surface projections (see *Figure 3B*), visual looming enhanced and suppressed auditory responses in adjacent patches consistent with neurophysiological results in non-human primates (*Kayser et al., 2008*). Critically, as illustrated in the raster plots, the response profile of a vertex to visual stimuli significantly predicted its laminar profile for crossmodal modulation (see *Figure 3B* right). In A1, vertices whose activations were only weakly suppressed by visual looming in deeper laminae showed a greater visual-induced response enhancement for audiovisual looming again in deeper laminae. These results suggest that visual looming influences activations in A1 mainly in deeper layers with similar patchy topographies in unisensory and audiovisual contexts.

By contrast, modality-specific attention affected auditory and visual evoked responses predominantly at the cortical surface (see significant linear term in *Table 3*). The large attentional BOLD-response effects in superficial laminae dovetail nicely with previous neurophysiological research, which located effects of modality-specific attention in supragranular layers of auditory cortices (*De Martino et al., 2015a*; *Lakatos et al., 2009*). Yet, because the GE-EPI BOLD response measured at superficial laminae includes contributions from deeper laminae (*Harel et al., 2006*; *Markuerkiaga et al., 2016*), we cannot unequivocally attribute the attentional influences at the cortical surface solely to superficial layers as neural origin. It is also possible that the increase in attentional BOLD-response effects towards the cortical surface may arise from neural effects across all cortical layers. Consistent with this conjecture, the BOLD-response profiles to auditory induced activations (*Figure 2—figure supplement 1A* left) and attentional modulation (*Figure 4Ai* left) in A1 and PT are quite similar.

Critically, however, the laminar profiles for attentional effects were distinct from those for multisensory (i.e. crossmodal) effects in auditory cortices. This dissociation in laminar profiles in the same territories cannot be explained by vascular effects that limit the laminar specificity of the BOLD response, but strongly implies that multisensory and attentional mechanisms regulate information flow in auditory cortices via partly distinct neural circuitries (*De Martino et al., 2015a*; *Lakatos et al., 2009*). Further, the constant laminar profiles for crossmodal influences in auditory cortices suggest that visual signals influence auditory cortices mainly via infragranular layers – a finding that converges with a recent neurophysiological study in mouse that likewise highlighted deep layer six as the key locus for visual influences on auditory cortex (*Morrill and Hasenstaub, 2018*). Anatomical studies in monkeys have previously suggested that visual inputs can influence auditory cortices via three routes (*Ghazanfar and Schroeder, 2006*; *Musacchia and Schroeder, 2009*; *Schroeder et al., 2003*; *Smiley and Falchier, 2009*): i. thalamic afferents, ii. direct connectivity between sensory areas (*Falchier et al., 2002*; *Rockland and Ojima, 2003*) and iii. connections from higher order association cortices.

## Common laminar profiles, but distinct activation patterns for multisensory and attentional influences in visual cortices

In visual cortices, visual and auditory stimuli evoked responses that were maximal in absolute amplitude at the cortical surface, yet differed in their activation pattern. Visual looming stimuli induced activations in areas representing the centre of the visual field and deactivations in surrounding areas representing the periphery (*Shmuel et al., 2002*). In contrast to this visual 'centre-surround' pattern, auditory stimuli induced widespread deactivations (*Laurienti et al., 2002*; *Leitão et al., 2013*; *Mozolic et al., 2008*) particularly in peripheral visual field representations known to have denser direct fibre connections to auditory cortices (*Falchier et al., 2002*; *Rockland and Ojima, 2003*).

Thus, our study revealed three types of deactivations. In visual cortices, both visual and auditory stimuli induced deactivations that were most evident at the cortical surface, as previously reported for deactivations to ipsilateral visual stimuli (*Fracasso et al., 2018*), though see [*Goense et al., 2012*]. In auditory cortices, the visual-induced deactivations were constant across cortical depth. These differences in laminar profiles or BOLD patterns between visual and auditory cortices may reflect differences in neural mechanisms. For instance, neurophysiological studies in rodents have shown asymmetries in multisensory influences between auditory and visual cortices, reporting robust auditory induced inhibition in layers 2/3 in visual cortices but not visual-induced inhibition in auditory cortices (*Iurilli et al., 2012*).

While we observed no statistically significant crossmodal nor attentional effects on the regional BOLD response profiles in visual cortices, multivariate pattern analyses indicated that modality-specific attention altered the activation patterns at a sub-regional spatial resolution (*Figure 4Aii*): attention to vision amplified activations in areas representing the central visual field and suppressed activations in areas representing peripheral visual fields (see *Figure 4Bii*).

The differences in activation patterns for multisensory and attentional influences indicate that the impact of purely auditory looming (*Maier et al., 2008*; *Maier et al., 2004*) on visual cortices cannot solely be attributed to a withdrawal of attentional resources from vision. Instead, purely auditory looming may inhibit activations in visual cortices via direct connectivity between auditory and visual areas (*Bizley et al., 2007*; *Budinger et al., 2006*; *Campi et al., 2010*; *Falchier et al., 2002*; *Ibrahim et al., 2016*; *Rockland and Ojima, 2003*). In line with this conjecture, research in rodents has shown that auditory stimuli modulates activity in supragranular layers in primary visual areas via direct connectivity from auditory cortices and translaminar inhibitory circuits – though the studies disagreed in whether layer 5 (*Iurilli et al., 2012*) or layer 1 (*Deneux et al., 2019*; *Ibrahim et al., 2016*) were the primary targets of auditory inputs.

## Multisensory effects in visual and auditory cortices – a comparison

Our results reveal distinct laminar profiles for multisensory deactivations in auditory and visual cortices. In auditory cortices the visual induced deactivations were constant across laminae, in visual cortices the deactivations linear increased (in absolute magnitude) toward cortical surface. These distinct laminar profiles converge with previous research in rodents that also reveal visual influences in auditory cortices predominantly in infragranular layer (*Morrill and Hasenstaub, 2018*), but auditory influences in visual cortices mainly in supragranular layers (*Deneux et al., 2019*; *Ibrahim et al., 2016*; *Iurilli et al., 2012*). Potentially, these distinct laminar profiles in auditory and visual cortices may suggest that multisensory influences are mediated by different neural circuitries in auditory and visual cortices. However, the distinct laminar profiles across cortical areas may not only reflect differences in neural circuitries but also in vascular organization. Moreover, a wealth of research has previously shown that multisensory responses profoundly depend on stimulus salience and other input characteristics (*Deneux et al., 2019*; *Meijer et al., 2017*; *Stein et al., 2020*; *Werner and Noppeney, 2010*). It is currently unclear how these factors influence laminar BOLD-response profiles. Finally, interpreting laminar BOLD-response profiles in relation to previous findings in rodents remains tentative because of cross-species and methodological differences.

## Conclusions

Using submillimetre resolution fMRI at 7T, we resolved BOLD response and activation patterns of multisensory and attentional influences across cortical depth in early sensory cortices. In visual cortices auditory stimulation induced widespread inhibition, while attention to vision enhanced central but suppressed peripheral field representations. In auditory cortices, competitive and cooperative multisensory interactions were stronger deep within the cortex, while attentional influences were greatest at the cortical surface. The distinctiveness of these depth-dependent activation profiles and patterns suggests that multisensory and attentional mechanisms control information flow via partly distinct neural circuitries.

## Materials and methods

All procedures were approved by the Ethics Committee of the University of Leipzig.

## Participants

Thirteen healthy German native speakers (6 females; 7 males; mean age: 24.8 years, standard deviation: 1.5 years, range: 22–27 years) gave written informed consent to participate in this fMRI study. Participants reported no history of psychiatric or neurological disorders, and no current use of any psychoactive medications. All had normal or corrected to normal vision and reported normal hearing. Two of those subjects did not complete all fMRI sessions and were therefore not included in the analysis. All procedures were approved by the Ethics Committee of the University of Leipzig.

## Looming stimuli and targets

The visual looming stimulus (500 ms duration) was a sequence of 1 to 4 radially expanding white annuli (*Figure 1B*). Each of the four annuli started in the centre of the screen with a diameter of 0.25° visual angle and expanded at a radial speed of 15.52° visual angle per second to a maximum radius of 3.88° visual angle which covered the entire screen visible in the scanner bore along the vertical dimension. During the 500 ms sequence, an additional annulus started in the centre every 125 ms. Given the retinotopic organization of early visual cortices (*Wandell et al., 2007*) these parameters ensured that visual looming elicited relatively widespread activations in visual cortices.

The auditory looming stimulus (500 ms duration) was composed of several pure tones (55, 110, 150, 220, 330, 380, 440, 660, 880, 990, 1110, 1500, 2500, 3000 Hz) that increased in frequency and amplitude over the entire 500 ms duration. The amplitude of the sound was exponentially modulated with the amplitude at time t given by $A_t = A_0 e^{0.68t}$-1 and $A_0$ being the initial baseline amplitude for each pure tone. The frequency of each pure tone component was increased by two thirds of its original base frequency over the 500 ms. Given the tonotopic organization of primary auditory cortices (*Formisano et al., 2003*) this mixture of sound frequencies ensured widespread activations in auditory cortices.

Audiovisual stimuli were generated by presenting the auditory and visual stimuli in synchrony together.

The visual target was a grey dot presented for 50 ms within a circular area that increased over the 500 ms duration consistent with the auditory, visual or audiovisual looming stimulus (*Figure 1B*). Auditory targets were a single 440 Hz pure tone that lasted for 100 ms. The size of the visual target and the sound amplitude of the auditory target were adjusted in a subject-specific fashion to match the approximate target detection performance across sensory modalities and then held constant across the entire experiment. This was done using the method of constant stimuli in a brief 'psychometric function experiment', performed prior to the fMRI study inside the scanner and with scanner noise present.

## Experimental design

The experiment conformed to a 3 (stimulus modality: auditory, visual, audiovisual) X 2 (attended modality: auditory, visual) factorial design (*Figure 1A*). In an intersensory selective attention paradigm, participants were presented with blocks of audio (A), visual (V) or audiovisual (AV) looming stimuli as highly salient stimuli that drive bottom-up attention and elicit crossmodal deactivations (*Leitão et al., 2013*). Brief auditory (i.e. weak and brief beep) and visual (i.e. small and brief grey dot) targets were presented interspersed in all stimulation blocks irrespective of the sensory modality of the looming stimuli or the focus of modality-specific attention (e.g. A and V targets were presented during a unisensory auditory looming block and auditory attention). The visual grey dot could be presented at any time at any location within the area defined by the radially expanding outermost annulus until they reach a radius of maximal 3.88° visual angle. Likewise, the auditory target is presented at random time interspersed in the looming sound. Consistent with previous research on modality-specific attention (*Lakatos et al., 2009*; *Werner and Noppeney, 2011*; *Werner and Noppeney, 2010*) these additional targets were included to ensure that the effects of attention could not be attributed to top-down effects associated with explicit responses.

During auditory attention ($Att_A$), participants were instructed to respond selectively to all auditory targets and ignore visual ones. During visual attention ($Att_V$) participants were instructed to respond selectively to all visual targets, that is brief grey dots and ignore auditory targets. In this way, we matched the auditory (i.e. attention to frequency) and visual attention (i.e. attention to targets in

spatial field) tasks to the tonotopic and retinotopic organizational principles of primary auditory and visual areas as well as the specifics of auditory and visual looming bottom-up salient stimuli.

## Experimental procedures

Auditory, visual or audiovisual looming stimuli were presented in 33 s blocks of 50 stimuli (stimulus duration: 500 ms, inter-stimuli interval: 160 ms giving a fixed stimulus onset asynchrony of 660 ms with no jitter) (*Figure 1B*). Activation blocks were preceded by 16 s fixation baseline periods. We opted for a block design as they have the highest design efficiency despite their potential drawbacks from a cognitive perspective (*Friston et al., 1999*). Throughout the entire experiment, participants fixated a cross (0.16° visual angle) in the centre of the black screen. Therefore, processes related to fixation are present in all conditions (i.e. all stimulation and fixation conditions). The fixation cross was white during the activation conditions and blue during the fixation baseline periods. A blue letter ('V' or 'A') presented 3 s prior to the activation blocks instructed participants to direct their attention to the visual or auditory modality. During visual attention blocks, participants were instructed to attend generally to the visual modality and respond selectively to visual targets. Conversely, during auditory attention blocks, participants had to attend generally to the auditory modality and respond selectively to auditory targets. Sixteen targets of each modality were presented for each condition. To maintain observer's attention and minimize the effect of target stimuli on measured stimulus-induced activations, targets were more frequently presented at the end of the block (i.e. two thirds of the blocks included one auditory target and one visual target, the last third of blocks included 2 targets of each type). The final block of each run was followed by a fixation baseline period of 20 s. The order of blocks was pseudo randomized across participants. Each fMRI run included 18 blocks (three blocks for each condition in our 2 × 3 design) and lasted 15 min. There were four runs per participant yielding 12 blocks per condition per participant.

## Experimental set up

Stimuli were presented using Presentation (v17.6, Neurobehavioral system, www.neurobs.com; RRID:SCR_002521) on a PC desktop (Windows XP). Visual stimuli were back-projected onto a plexiglas screen using an LCD projector (60 Hz) visible to the participant through a mirror mounted on the MR head coil. The MR head coil enabled a visual field with a width of 10.35° and height of 7.76° visual angle. Auditory stimuli were presented using MR-compatible headphones (MR-Confon; HP-SI01 adapted for the head-coil). Behavioral responses were collected using a MR-compatible button device connected to the stimulus computer. The code to run the fMRI experiment is available at https://doi.org/10.5281/zenodo.3581316.

## Behavioral data analysis

The response window extended from the target onset and for either 2.5 s or until the beginning of the next target. Any recorded response outside of that window was coded as an extra-response (i.e. to be modeled in the fMRI design matrix). The percentage of auditory (or visual) targets that participants responded to in a particular condition were entered as dependent variables into a 2 (target modality: auditory vs. visual) X 2 (attended modality: auditory vs. visual) X 3 (stimulus block modality: auditory, visual, audio-visual) repeated measures ANOVA (results were corrected for non-sphericity using Greenhouse-Geiser correction). Alpha was set to 0.05. This analysis was done with IBM SPSS Statistics, version 17.0 (IBM, Armonk, NY, USA).

## MRI data acquisition

All experiments were performed on a Siemens 7T MAGNETOM MRI scanner (Siemens Healthineers, Erlangen, Germany) equipped with an SC72 gradient coil (Siemens Healthineers). For radio-frequency transmission and reception, a single-channel-transmit/32-channel-receive phased array head coil (Nova Medical, Wilmington, MA) was used. The experiment was run over 2 days with 50 min MRI scanning in total per day per participant.

On the first day a whole-brain MP2RAGE sequence (*Marques et al., 2010*) was used to acquire a quantitative T1 map and a weighted T1 image and a second inversion image to enable high-resolution segmentation and cortical laminae definition [repetition time (TR)/echo time (TE) = 5000/2.45 ms, inversion time 1 = 900 ms, flip angle 1 = 5°, inversion time 2 = 2750 ms, flip angle 2 = 3°, 240

slices, matrix = 320 × 320, acquisition time = 600 s, spatial resolution = 0.7 × 0.7 × 0.7 mm³ voxels, partial Fourier factor = 6/8, parallel imaging using GRAPPA with an acceleration factor of 4, field of view = 168×224 mm]. On the second day, we used a shorter version of that sequence with a coarser resolution only in order to position our EPI acquisition slab.

Two functional runs were acquired every day using a 2D gradient echo single-shot echo planar imaging (EPI) readout [TR/TE = 3000/25 ms, flip angle = 90°, 48 axial slices acquired in descending direction, matrix size = 256 × 240, phase encoding along the second dimension (posterior to anterior), slice thickness = 0.75 mm, pixel bandwidth: 1085 Hz, brain coverage = 36 mm, spatial resolution = 0.75 × 0.75 × 0.75 mm³ voxels, partial Fourier factor = 6/8, parallel imaging using GRAPPA with an acceleration factor of 4, number of volumes = 302, acquisition time = 906 s, field of view = 192×180 mm. In order to avoid spin history effects due to imperfect pulse profiles a slice gap of 0.05 mm was introduced. For every session, a new and optimum set of shim values was used.]. A corresponding vendor-provided homodyne online image reconstruction algorithm was used. The first six volumes in each run were automatically discarded to avoid for T1 saturation effects.

The acquisition slab was inclined and positioned to include Heschl's gyri, the posterior portions of superior temporal gyri and the calcarine sulci. Both hemispheres were acquired. On the second day the slab was auto-aligned on the acquisition slab of the first day. This provided us with nearly full coverage over all our four regions of interest (see *Supplementary file 2*).

One field-map was acquired on each day with a coverage and orientation matching that of the EPI slab [TR/TE1/TE2 = 1500/6/7.02 ms, flip angle = 72°; 18 axial slices, matrix = 96×90, slice thickness = 2 mm, spatial resolution = 2.00×2.00 X 2.00 mm³ voxels, field of view: FOV = 192×180 mm].

## MRI structural analysis

The MRI structural data were processed using the CBS toolbox (v3.0.8, Max Planck Institute for human cognitive and brain science, Leipzig, Germany, www.nitrc.org/projects/cbs-tools/; RRID:SCR_009452) based on the MIPAV software package (v7.0.1, NIH, Bethesda, USA, www.mipav.cit.nih.gov, (*McAuliffe et al., 2001*); RRID:SCR_007371) and the JIST pipeline environment(v2.0, (*Landman et al., 2013*; *Lucas et al., 2010*); RRID:SCR_008887). The T1 map was coregistered to the Montreal's Neurological Institute brain space (rigid body registration and normalized mutual information as a cost function) and resampled to an isometric resolution of 0.4 mm. The image was segmented fully automatically, and the cortical boundary surfaces were reconstructed (*Bazin et al., 2014*). Between these surfaces, we computed six intracortical surfaces that defined cortical laminae with the equivolume approach (*Waehnert et al., 2014*) (*Figure 1C* and *Figure 1—figure supplement 2*). It is important to note that the term 'lamina' used in this communication does not directly refer to cytoarchitectonically defined cortical layers: for example, the variation across cortical regions of the relative thickness between layers was not modelled here (*Nieuwenhuys, 2013*). However, it should be mentioned that this method provides biologically plausible cortical contours (*Waehnert et al., 2014*).

## Definition of region of interest

We defined four regions of interest (ROI), each combining left and right hemispheres, as follows (see Table S2 for ROI effective sizes):

A. Several studies have shown that the core of primary auditory cortex (A1) because of its characteristic myelination profile can be localized using its low T1 intensity (*De Martino et al., 2015b*; *Dick et al., 2012*; *Hackett et al., 2001*). We therefore defined it by manual delineation (RG, UN) that was guided by anatomical structure (Heschl's gyrus) and informed by the quantitative T1 map that was sampled at mid-depth of the cortex and projected on the corresponding inflated surface. The final bilateral A1 region was defined as the union of the two delineations (RG, UN). As a result the sub-region of Heschl's gyrus that we refer to as primary auditory cortex may differ from primary auditory cortex defined based on cytoarchitecture.

B. The planum temporale (PT) was defined bilaterally using the area 41–42 of the brainnetome atlas (*Fan et al., 2016*) (http://atlas.brainnetome.org/). The full probability maps of this ROI and its anatomical neighbours were inverse-normalized into native space using the deformation field given by the SPM segmentation of the T1 weighted structural scan. The probability map was then re-gridded to an isometric resolution of 0.4 mm, sampled at mid-depth of the

cortex and projected on the corresponding surface. Vertices were included as part of PT if their probability exceeded 40% unless i. they were already defined as being part of A1 or ii. the sum of the probabilities of the neighbouring regions exceeded that of the area 41–42.

C. Primary Visual area (V1) and high order visual areas (V2 and V3) were defined bilaterally in a similar fashion as PT. We used the probabilistic retinotopic maps of the ROIs for V1, V2/3 (*Wang et al., 2015*) (http://scholar.princeton.edu/napl/resources). In this case vertices were included if their probability exceeded 10% unless the sum of the probabilities of the neighbouring regions exceeded that of the area of interest.

We obtained >95% coverage for A1, PT and V1 and >82% coverage for V2/3. (For the number of vertices and percentage coverage across subjects, *Supplementary file 2*).

## fMRI preprocessing

The fMRI data were pre-processed and analyzed using statistical parametric mapping (SPM12 – v6685; Wellcome Center for Neuroimaging, London, UK; www.fil.ion.ucl.ac.uk/spm; RRID:SCR_007037) running on matlab (Mathworks). The fieldmaps were co-registered to the first functional scan of the first run (i.e. rigid-body transformations optimized using normalized mutual information as cost function) and a voxel displacement map was then created. Functional scans from each participant were realigned and unwarped using the first scan as a reference (interpolation: 4th degree b-spline, unwarping done using the voxel displacement map of the corresponding day for each run).

## Functional to anatomical coregistration

We co-registered the mean EPI image to the pre-processed T1 map (i.e. rigid-body transformations optimized using normalised mutual information as cost function). We assessed the accuracy of co-registration using FSLview (FSL 5.0 - Analysis Group, FMRIB, Oxford, UK, https://fsl.fmrib.ox.ac.uk/fsl/fslwiki/FSL; RRID:SCR_002823) by flipping back and forth between the mean EPI and the T1 map images in all the ROIs (see *Figure 1—figure supplement 2*). If the co-registration was not sufficiently precise, the mean EPI was initially co-registered manually and the co-registration repeated until anatomical structures of the mean EPI and the T1 map were precisely realigned. The transformation matrix of the co-registration was later applied to the beta images from the fMRI general linear model (see fMRI statistical analysis).

## fMRI statistical analysis

At the first (i.e subject) level, we performed a mass univariate analysis with a linear regression at each voxel, using generalized least squares with a global approximate AR(1) autocorrelation model and a drift fit with discrete cosine transform basis (128 s cut-off). We modelled the fMRI experiment with a mixed block-event related model with regressors entered into the run-specific design matrix after convolving the onsets of each block or event with a canonical hemodynamic response function (HRF) and its temporal derivative. More specifically, we modelled each of the three looming blocks in each run for each of the six conditions in our 3 (visual, auditory, audiovisual) X 2 (auditory vs. visual attention) factorial design separately. Hence for each run the statistical model included the following regressors: 3 block regressors for each of the 6 conditions (A_Att$_A$, A_Att$_V$, V_Att$_A$, V_Att$_V$, AV_Att$_A$, AV_Att$_V$), 4 event-related regressors for each target type under each attention condition (i.e. TargetA_Att$_A$, TargetA_Att$_V$, TargetV_Att$_A$, TargetV_Att$_v$) and one event-related regressor modelling any additional responses that participants produced in the absence of targets. We modelled the targets and additional responses as independent regressors in our first (i.e. subject) level general linear model (GLM) to minimize confounding effects of perceptual decision making, response selection and motor preparation on the activations reported for the A, V, AV looming activation blocks that are the focus of this communication. Nuisance covariates included the realignment parameters to account for residual motion artefacts.

All subsequent analyses included only the beta images pertaining to the HRF of the activation blocks in our 2 (modality specific attention) X 3 (stimulation modality) design: 3 blocks per condition per run X 6 conditions X 4 runs = 72 beta images per subject.

To minimize the possibility that attentional lapses reduced the sensitivity of our analysis we also repeated our neuroimaging analysis selectively for blocks that included neither misses (i.e. missed responses to targets of the attended modality) nor false alarms (i.e. responses to targets from the

unattended modality). This control analysis basically showed similar results similar to those from the main analysis that is reported in this manuscript.

## Sampling of the BOLD-response along cortical depth

The 72 beta images (6 conditions X 3 blocks per condition per run X 4 runs) were co-registered to the MRI structural by applying the transformation matrix obtained from the mean EPI image co-registration and up-sampled to a 0.4 mm isometric resolution (4th degree b-spline interpolation). Finally, we sampled each beta image along vertices defined by the normal to the mid-cortical surface at the depths defined by the 6 intra-cortical surfaces from the MRI structural analysis (tri-linear interpolation) (see 'MRI structural analysis' above, *Figure 1C*).

In total, the fMRI data were therefore resampled 3 times for: 1. realignment+unwarping, 2. upsampling the beta images to 0.4 mm and 3. sampling the beta images along the surfaces. The smoothness estimation using AFNI 3dFWHMx gave the following results (FWHM mean and standard deviation across subjects in X/Y/Z): smoothness of the raw data: 1.70 (0.08), 2.05 (0.09), 1.65 (0.08) mm; smoothness after upsampling at 0.4 mm: 2.22 (0.14), 2.88 (0.31), 2.11 (0.21) mm. We were not able to estimate the smoothness of the data after sampling along the surfaces. Note that no additional smoothing was applied beyond the one due to interpolation during pre-processing.

The activity values at the vertices of these 6 intra-cortical surfaces in our four ROIs (i.e. primary auditory, planum temporale, V1, V2/3) pooled over both hemispheres form the basis for our univariate ROI analysis of the laminar BOLD-response profiles and laminar decoding accuracy profiles (based on multivariate pattern classification).

## Laminar BOLD-response profiles for contrasts of interest

The laminar profiles of each ROI were obtained for each block per run by collapsing data over vertices. For each of our 72 beta images (6 conditions X 3 blocks per condition per run X 4 runs) we summarized the BOLD response at each of the six cortical depth levels in terms of the median of the parameter estimates across all vertices at this cortical depth within a particular ROI (n.b. each ROI pools over both hemispheres). The median was used as a summary index because the parameter estimates distribution was skewed especially for the most superficial laminae.

For each ROI and lamina, we computed the following contrasts of interest independently for the $i^{th}$ block of a given condition of the $j^{th}$ run:

1. The deactivation induced

    a. by auditory stimuli relative to fixation baseline irrespective of modality-specific attention: $[A - \mathrm{Fix}] = [A\_Att_A + A\_Att_V]i, j/2$,
    b. by visual stimuli relative to fixation baseline irrespective of modality-specific attention: $[V - \mathrm{Fix}] = [V\_Att_A + V\_Att_V]i, j/2$,
2. The crossmodal enhancement or suppression irrespective of attention, i.e. specifically for auditory and visual regions:
    a. A1 and PT: the visual-induced modulation of auditory activations irrespective of modality-specific attention: $[AV - A] = [(AV\_Att_A - A\_Att_A) + (AV\_Att_V - A\_Att_V)]i, j/2$,
    b. In V1 and V2/3: the auditory-induced modulation of visual activations irrespective of modality-specific attention: $[AV - V] = [(AV\_Att_A - V\_Att_A) + (AV\_Att_V - V\_Att_V)]i, j/2$,
3. Attentional modulation irrespective of stimulus modality, that is specifically for auditory and visual regions

    a. A1 and PT: Modulation of stimulus responses by auditory relative to visual attention $[Att_A - Att_V]_{A,V,AV} = [(Att_A - Att_V)_A + (Att_A - Att_V)_V + (Att_A - Att_V)_{AV}]i, j/3$.
    b. V1 and V2/3: Modulation of stimulus responses by visual relative to auditory attention $[Att_V - Att_A]_{A,V,AV} = [(Att_V - Att_A)_A + (Att_V - Att_A)_V + (Att_V - Att_A)_{AV}]_{i,j}/3$

For completeness, we also assessed whether the differences in BOLD-response for the contrasts listed 1 and 2 – in cases when they were significantly different from zero - depended on modality-specific attention (i.e. by testing for the interaction of sensory evoked responses or crossmodal enhancement with modality-specific attention). Moreover, we also show the activations > fixation for auditory or visual stimuli in the *Figure 2—figure supplement 1* to confirm that our experimental manipulations were effective.

## Shape parameters for characterization of laminar BOLD response profiles

The laminar profile of each ROI was then summarized by parameters by collapsing across blocks per runs. For each statistical comparison, lamina and ROI, we thus obtained 12 (i.e. 3 blocks X 4 runs) contrast estimates (as the median over vertices within a bilateral ROI) per subject that is 12 laminar profiles. We refer to those parameter estimates that quantify the BOLD response for a particular lamina as BOLD response parameter estimates. For instance, *Figure 2A* (first row) shows the. B parameter estimates (across participants mean ± SEM) as a line plot for each of the six laminae as a function of cortical depth along the x-axis.

To characterize the overall shape of the laminar profile, we estimated for each subject a second-level general linear model (i.e. a laminar GLM) that modeled the activation in each lamina in a given ROI as dependent variable by two predictors: 1. a constant term (i.e. activation mean across laminae) and 2. a linear term characterizing a linear increase across laminae (mean-centered and orthogonalized with respect to the constant term). The parameter estimates of this 2nd or laminar GLM are referred to as shape parameters. For instance, *Figure 2A* (2nd row) shows the S parameter estimates as violin plots for the constant and the linear term. The S-parameters enable us to make inferences about the shape of a laminar BOLD-response profile rather than using an omnibus F-test that assesses whether the BOLD-response differs across any of the six laminae followed by numerous post hoc pairwise comparisons between layers. However, we acknowledge that this characterization makes our inference less specific about locating even the BOLD-response to a particular lamina. Moreover, because the laminar GLM in the current report did not include any higher order (e.g quadratic) terms they would not be to model U-shaped laminar profiles.

## Statistical analysis for the shape parameters at the between subject that is group level

To enable generalization to the population, we entered these shape parameters into linear mixed effects models at the group level separately for each contrast. To limit the number of statistical tests we used the following step-down approach:

1. We formed 2 (shape parameter: constant, linear) x 2 (ROI: primary, non-primary) linear mixed effects models separately for the six contrasts, that is 2 (sensory cortices: visual, auditory) x 3 (contrasts: crossmodal deactivation, crossmodal modulation, attentional modulation). In each linear mixed effects model we tested whether the constant or the linear parameter, each averaged across primary and higher order ROIs, was significantly different from zero using a two dimensional F-contrast. Hence, we computed three F-tests for visual cortices and three F-tests for auditory cortices.
2. If a two dimensional F-test was significant, we computed follow-up one dimensional F-tests separately for the constant and the linear parameters (again averaged across primary and higher order sensory cortices).
3. If a one dimensional F-test was significant, we computed follow-up t-tests separately for the primary and the higher order sensory cortices.

Based on our a-priori predictions (*Leitão et al., 2013*) that auditory stimuli induce deactivations in visual areas and that visual stimuli induce deactivations in auditory areas, we employed one-sided t-tests in step 2 and 3 (i.e. negative constant term for contrasts). In all other cases unidirectional F-tests were used. Unless otherwise stated, we report statistical results as significant at p<0.05.

## Multivariate analysis of pattern across vertices and decoding accuracy profiles

Multivariate pattern analyses were performed using a linear support vector classifier (SVC) that was trained in a leave-one-run-out cross-validation scheme (LIBSVM 3.21, https://www.csie.ntu.edu.tw/~cjlin/libsvm/; RRID:SCR_010243, with C = 1 and mean-centred activations for each feature separately for training and test sets).

We performed multivariate pattern analysis directly on the BOLD response patterns pooled over the two hemispheres, that is the patterns of B-parameters, independently at each of the six cortical depths to generate laminar profiles of decoding accuracy for the following comparisons:

1. [AV vs A]$_{AttA, AttV}$ (pooled over attended modality) for A1 and PT,

2. [AV vs V]$_{AttA, AttV}$ (pooled over attended modality) for V1 and V2-3,
3. [Att$_A$ vs Att$_V$]$_{A,V,AV}$ (pooled over stimulation modality) for A1, PT, V1 and V2/3.

For completeness, we also assessed whether the classification performance for the comparisons listed above – in cases when they were significantly different from zero - depended on modality-specific attention by running the comparison for the equivalent contrast between the the two modality specific attention conditions (e.g the comparison [AV-V]$_{att\ A}$ vs [AV-V]$_{att\ V}$ for the one in the list above).

In line with our analysis of laminar BOLD response profiles, we modeled the laminar profiles of decoding accuracy using a laminar GLM with a constant term and a linear term as linear increase across laminae as predictors.

Again as in our analysis of laminar BOLD-response profiles, the 'constant' and 'linear' shape parameters characterizing the decoding accuracy profiles were entered into 2 (shape parameter: constant, linear) x 2 (ROI: primary, non-primary) linear mixed effects models separately for each of the four decoding comparisons, that is 2 (sensory cortices: visual, auditory) x 2 (contrasts: crossmodal modulation, attentional modulation). We then applied the step down procedure exactly as described in detail for the BOLD-response profiles. Please note that the laminar profiles of decoding accuracy need to be interpreted cautiously. First, decoding accuracy depends on BOLD-signal and noise characteristics that can both vary across laminae. Second, the laminar profile of decoding accuracy is more difficult to interpret, because accuracy is bounded between 0 and 1.

Raster plots: statistical relationship of BOLD response profiles between different conditions.

Next, we explored whether the response profile in a vertex as characterized by their 'constant' and 'linear' shape parameters in one condition is statistically predictive of this vertex laminar profile in another condition or contrast (see figure 5 from [*Fracasso et al., 2018*]. For instance, we asked whether the magnitude of visual induced deactivations (e.g. [V-Fix] averaged over attention conditions) in a vertex predicts its crossmodal enhancement (e.g. [AV-A] averaged over attention conditions). For this, we sorted and averaged the vertices in percentile-like bins based on the value of a shape parameter (e.g. 'constant' or 'linear') for a particular sorting contrast (e.g. [V–Fix]).

This bin order was then used to sort and average the values of the 2nd contrast (e.g. [AV-A]). We entered the shape parameter value (e.g. 'constant' or 'linear') for the sorting contrast (e.g. [V–Fix]) for each bin as a linear regressor (+ a constant) to predict the corresponding shape parameter in each bin in the sorted contrast (e.g. [AV-A]) in a subject-specific general linear model.

To allow for generalization to the population, we entered the parameter for the linear term (i.e. slope of regression line) of this regression model into one sample t-tests at the group level.

We illustrated the statistical relationships of the laminar profiles between different contrasts at the group level in raster plots by averaging the laminar profiles for each bin across subjects of the sorting (i.e. predicting) and the sorted (i.e. predicted) values. For instance, in *Figure 3B* i (right) we sorted the vertices according to [V-fix] in PT such that the constant parameter (i.e. average BOLD response across laminae) increases from bottom to top (i.e. predicting contrast). Unsurprisingly, the raster plot for the [V-Fix] panel as the predicting contrast thus goes from a negative BOLD response profile (i.e. coded in blue) in the bottom rows of the panel to a positive BOLD response profile (i.e. coded in red) in the top rows of the panel. Under the null-hypothesis the laminar BOLD response profile for the predicting contrast (e.g. [V-fix]) in a vertex is unrelated to its laminar BOLD response profile for the predicted contrast (e.g. [AV-A]) and we would not expect any structure in the raster plots for the predicted contrast (e.g. [AV-A], n.b. we accounted for spurious correlation by performing all these analyses in a crossvalidated fashion). Conversely, if the shape parameter for [V-fix] in a vertex significantly predicts its shape parameters for [AV-A], we would expect a structured raster plot also for [AV-A].

Hence, these raster plots can reveal additional BOLD signal structure that is averaged out by the group surface projections: while group surface projections (see next section, or *Figure 3B* left) reveal only spatial topographies which are consistent across participants, the raster plots reveal whether the similarity between patterns of laminar profiles from different conditions is consistent across subjects, even when the activation patterns themselves are not similar across subjects. In other words, raster plots illustrate the similarity or covariance between patterns that is consistent across subjects even when the patterns themselves vary across subjects.

Please note that both the regression model and the raster plots were computed in a leave one day out cross-validation to avoid biases and spurious correlations between sorting and sorted contrast. The number of bins was set to the smallest number of vertices found within an ROI (pooled over hemispheres) across subjects. For visualization purposes all depicted raster plots were smoothed along the vertical axis (FWHM = 1% of the number of data bins).

We also note that the statistical results were basically equivalent without any binning (i.e. directly entering the shape parameter values of vertices into the GLM), the binning was applied only to enable illustration of the results in raster plots.

These regression models over vertices and raster plots were used to evaluate whether crossmodal modulation for auditory or visual stimuli in A1, PT, V1 or V2/3 depended on the unisensory visual or auditory response.

### Intersubject registration for surface projection of group results

To visualize patterns of the laminar profile shape parameters (i.e. 'constant' or 'linear') that are consistent across subjects on the cortical surface, we transformed the individual surface projections into a standardized study group space using a multimodal, multi-contrast surface registration (MMSR) approach (*Tardif et al., 2015*). The transformation matrix for group normalization was computed for the level-set corresponding to the mid-cortical surface and the high-resolution T1 maps. The T1 map was sampled radially along the cortical profile, averaged between 20% and 80% cortical depth and smoothed tangentially (FWHM = 1.5 mm). To reduce computation time, the level-set and the T1 map data were down-sampled to a resolution of 0.8 mm isotropic. We used a two-stage registration. In the first step, we computed an intermediate mean surface by using the median subject in the study group as the initial target. In the second step, MMSR was repeated using this intermediate mean surface as the new target.

To apply this registration to our fMRI data, we computed the mean beta images for each of our six conditions, sampled them at the depths defined by the six intra-cortical surfaces (trilinear interpolation) and then down-sampled to a resolution of 0.8 mm isotropic. We then used the deformation field resulting from the second stage MMSR to transform those 36 images per subject (six conditions X six depths) into the group surface template space (trilinear interpolation). Mass univariate laminar GLM were then performed in normalized group space and the parameters corresponding to the 'constant' and 'linear' terms were averaged across subjects for each vertex.

### Materials and data availability

The raw data of the results presented here are available in a BIDS format upon request (remi.gau@gmail.com). Beta values extracted from our layers/ROIs for each participant as well as the summary data necessary to reproduce our figures have been uploaded as CSV or mat files on the open-science framework (https://osf.io/63dba/). Group average statistical maps are available in an NIDM format from neurovault (https://neurovault.org/collections/5209/).

The results of the quality control MRIQC pipeline (https://mriqc.readthedocs.io/en/stable/) on the BOLD data as well as information about motion and framewise displacement during scanning is also available from the repository.

The code for the analysis is available at https://doi.org/10.5281/zenodo.3581319. The code to run the fMRI experiment is available at https://doi.org/10.5281/zenodo.3581316.

## Acknowledgements

This project was funded by the ERC starter grant (mult-sens) and the Max Planck Society.

## Additional information

### Funding

| Funder | Grant reference number | Author |
|---|---|---|
| European Research Council | Mult-sens | Uta Noppeney |
| Max Planck Society | | Robert Turner |

The funders had no role in study design, data collection and interpretation, or the decision to submit the work for publication.

## Author contributions
Remi Gau, Resources, Methodology, Writing - review and editing; Pierre-Louis Bazin, Resources, Software, Writing - review and editing; Robert Trampel, Resources, Methodology; Robert Turner, Conceptualization, Resources, Supervision, Funding acquisition, Methodology, Project administration, Writing - review and editing; Uta Noppeney, Conceptualization, Resources, Formal analysis, Supervision, Funding acquisition, Validation, Methodology, Writing - original draft, Project administration, Writing - review and editing

## Author ORCIDs
Remi Gau (ID) https://orcid.org/0000-0002-1535-9767
Pierre-Louis Bazin (ID) https://orcid.org/0000-0002-0141-5510

## Ethics
Human subjects: All procedures were approved by the Ethics Committee of the University of Leipzig under the protocol number 273-14: "Magnetresonanz-Untersuchungen am Menschen bei 7 Tesla". Participants gave written informed consent to participate in this fMRI study.

## Decision letter and Author response
Decision letter https://doi.org/10.7554/eLife.46856.sa1
Author response https://doi.org/10.7554/eLife.46856.sa2

# Additional files

## Supplementary files
• Supplementary file 1. Behavioural results. Notes: Percentage of target responses (mean and STD across subjects) in the six conditions of our 2 × 3 experimental design. n = 11 Please note that responses to visual targets under auditory attention and auditory targets under visual attention are false alarms.

• Supplementary file 2. ROI size and coverage. Notes: 'Number of vertices' refers to the vertices with valid data at all the sampled cortical depths. This vertex count was divided by the total number of vertices included in the initial ROI definition to compute the 'Fraction of the ROI covered'. Note that those numbers are pooled over both hemispheres. n = 11

• Supplementary file 3. Auditory and visual activations. Using 2 (shape parameter: constant, linear) x 2 (ROI: primary, non-primary) linear mixed effects models, we performed the following statistical comparisons in a 'step down procedure':

• Supplementary file 4. Cross-modal modulation in visual areas. Using 2 (shape parameter: constant, linear) x 2 (ROI: primary, non-primary) linear mixed effects models, we performed the following statistical comparisons in a 'step down procedure':

• Transparent reporting form

## Data availability
Data (sufficient to recreate figures) are publicly available on the OSF project of this study: https://osf.io/63dba/. The raw data of the results presented here are available in a BIDS format upon request: the consent form originally signed by the participants did not allow for making raw data publicly available.

The following dataset was generated:

| Author(s) | Year | Dataset title | Dataset URL | Database and Identifier |
| --- | --- | --- | --- | --- |
| Gau R | 2019 | AV - attention - 7T | https://osf.io/63dba/ | Open Science |

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
