## [Decision Letter]

**Acceptance summary:**

This work uses high-field imaging and innovative analytical approaches to examine the laminar profile of two distinct modulatory influences on sensory processing: multisensory interactions and attentional modulation. Interestingly, multisensory and attentional mechanisms modulated the laminar activity profile in distinct ways. This strongly suggests that these two forms of modulatory responses modulate sensory processing via different neural pathways.

**Decision letter after peer review:**

Thank you for submitting your article "Resolving multisensory and attentional influences across cortical depth in sensory cortices" for consideration by *eLife*. Your article has been reviewed by three peer reviewers, including Floris P Lange as the Reviewing Editor and Reviewer #1, and the evaluation has been overseen by Christian Büchel as the Senior Editor. The following individuals involved in review of your submission have agreed to reveal their identity: Laurentius Huber (Reviewer #2); David G Norris (Reviewer #3).

The reviewers have discussed the reviews with one another and the Reviewing Editor has drafted this decision to help you prepare a revised submission.

Summary:

The manuscript entitled "Resolving multisensory and attentional influences across cortical depth in sensory cortices" describes a layer-fMRI study with audio and visual stimuli and it investigates layer dependent signal changes across different attention and modality conditions. The main conclusion of the study is that cross-modal activity modulates the deeper layers, whereas attentional differences modulate the superficial layers. The fact that this can be measured noninvasively in humans, will be of great interest to a large research field.

Essential revisions:

Some of the most significant concerns raised are:

1) Risk of false positive significance scores. Possibly exacerbated by a low sample size (N=11)?

2) None of the effects in the main conclusion are reproduced in control task conditions: The attention effect in visual areas is not reproduced by attention effects in auditory areas. The cross-modal modulation in auditory areas is not reproduced in visual areas (neither in size, direction, nor shape of layer signals).

3) Similarly, none of the effects in the main conclusion are reproduced across the alternative control analysis approaches (shape parameters vs. decodability): Crossmodal layer-profiles in A1 and V1 (Figure 3A) look very different for B-parameters compared to the respective decodability values (zero, constant, decreasing, zero).

These and other points are elaborated in the individual reviews below.

Reviewer #1:

The authors present an elegant study of visual, auditory and multimodal processing of looming stimuli. In addition, they employ an attention modulation, effectively generating a 3x2 design. They record high resolution (0.75mm isotropic) gradient-echo BOLD of 11 subjects and analyze it using the GLM framework. They use equivolume layering and the median activation over four ROIs (two auditory and two visual) to extract 4x6 laminar profiles. Summarizing their findings, they generally find the expected increase in visual areas for visual and auditory areas for auditory stimuli. They in addition find deactivations across modalities. They find cross-modal modulations only in auditory cortex, interestingly in Heschel's gyrus only in multi-voxel decoding, not in amplitude. Attentional modulations were mostly constrained to auditory regions as well.

The study is complex, but with a solid design. The analyses are described in detail, the code is freely available (the data not yet, so I didn't test their pipeline), and the methods used are appropriate. The paper is also well written, clear and consistent. The figures are detailed, give single subject estimates and are mostly clear. I especially enjoyed that the authors provide all results, not only the significant tests. This is even more important, given that they test quite a lot (which should probably be discussed a bit more, especially in light of a small n=11), and thus there is no hard significance-filter for the results.

Reviewer #2:

The manuscript entitled "Resolving multisensory and attentional influences across cortical depth in sensory cortices" describes a layer-fMRI study with audio and visual stimuli and it investigates layer dependent signal changes across different attention and modality conditions. The main conclusion of the study is that cross-modal activity modulates the deeper layers, whereas attentional differences modulate the superficial layers. The fact that this can be measured noninvasively in humans, will be of great interest to a large research field. And upon some revisions, I ultimately recommend its publication with great enthusiasm.

The novelty and strengths:

– While some other groups are also currently working on it, I believe this is the first manuscript of layer-dependent analyses of multi-modal integration.

– I believe I have not seen any layer-fMRI manuscript that combines so many different brain areas (including PT, which is new for layer-fMRI) and so many different task conditions in one study.

– The authors developed a novel analysis methodology of interpreting the layer-profiles as a combination of linear 'slopes' and 'constant' offsets that allow straightforward summary statistical tests across task conditions.

– The data-acquisition methodology is technically sound and appropriate to address the research questions. Like in previous studies of that group, they use the most advanced imaging hardware, sequences and imaging protocols. Without being too advanced that it would require additional method-validation-studies.

– The statistical results are shown very honestly in violin diagrams for all conditions in and all participants. And data will be shared.

The weaknesses:

– I feel the task design might have been pushed it a bit too much. There are as many as 6 task conditions with subtle differences. Thus, either one of the condition differentiations does not have so many trials to average across compared to comparable layer-fMRI studies. As a result, multiple different tasks conditions needed to be averaged together and the main conclusions are based on effects that (almost) disappear in the noise level.

– I believe the clarity of the manuscript can be improved. Each figure has up to 20 sub panels, whereas most of them show insignificant effects that cannot be used to support the main conclusion. It took me quite a while to filter out the relevant information.

– I believe the way the analysis is conducted and the data are presented could benefit from more discussions on the limits of their interpretability. I am hesitant whether I can interpret the shape parameters and decodability profiles as measures of neural activity in a way that the main conclusion is the most plausible explanation for all the depicted results.

Reviewer #3:

In this paper, the authors examine the responses of early sensory cortices to auditory, visual, and combined auditory-visual stimuli, in conjunction with an attentional modulation, using laminar resolution fMRI. The results presented are of interest, and convincing. Nevertheless, I feel that the paper could be improved by consideration of the following:

1) In the Abstract the claim is made that these findings are crucial for understanding "how the brain regulates information flow across senses". A sceptic could claim that the article only succeeds in reproducing animal literature, and I would suggest that the authors expand their Discussion so as to better substantiate this claim.

2) The figures are organised to present the results of auditory and visual stimulation separately, whereas the text is organised to first deal with auditory and then visual stimulation. It may be more logical to deal with both modalities in parallel. Beyond the criticism at the organisational level, the current structure rather masks obvious asymmetries between the responses in primary auditory and visual cortices. The narrative that emerges would seem to be of a dominance visual modality. The Discussion rather misses the opportunity to compare and contrast across the modalities. For example, there are obvious differences shown in Figures 2A, and 3A which could benefit from some in-depth discussion.

3) Cross modal modulation is assessed by examining the contrast AV-A in auditory cortex and correspondingly AV-V in visual cortex. To me it would seem more logical to construct a contrast AV-(A + V). For example, if AV-A would produce no significant activation, but a visual stimulus in isolation would deactivate auditory cortex, then there would be a clear cross-modal effect that would be missed by your chosen contrast parameter. I would suggest re-analysing the data.

4) Please justify the inclusion of, and discuss the interpretation of the results from, the planum temporale. In this context please also explain in the Results why the left hemisphere was imaged.

5) It could be easier to follow some of the text if the authors would deal with areas of early visual cortex that activate upon visual stimulus separately from those that deactivate upon visual stimulus.

6) I had some concerns about the stimulus design, which I could not find addressed in the text. First, the baseline condition of visual fixation is itself some form of visual task. Second, the auditory target apparently has a fixed amplitude but is presented at a variable time against a looming auditory stimulus: doesn't this affect the detectability?

7) The English is generally good, but the authors persistently introduce comparative statements with no clear object. This is particularly confusing when paragraphs start with: "by contrast"; "in contrast"; "hence". Most of these can be deleted without any effect on the meaning. The authors also tend to introduce a chronology ("next we…") which is not really necessary.

---

## [Author Response]

Essential revisions:Some of the most significant concerns raised are:1) Risk of false positive significance scores. Possibly exacerbated by a low sample size (N=11)?

Thanks. We appreciate the reviewer’s concern.

However, while we have indeed reported a large number of tests for full characterization of the data, our key hypotheses and research interests focused only on a subset of these contrasts:

1) Crossmodal deactivations: (i) A in visual cortices, (ii) V in auditory cortices

2) Crossmodal modulation: (iii) AV > V in visual cortices, (iv) AV > A in auditory cortices

3) Attentional modulation: (v) Att V vs. Att A in auditory cortices, (vi) Att A vs. Att V in visual cortices

Further, we discussed only those effects that were consistently observed in both primary and higher order areas (e.g. A1 and PT). This results in 6 statistical comparisons for the mean BOLD-response. Further, we also assessed decoding accuracy for the crossmodal modulation and attentional modulation giving a total of 10 statistical comparisons.

We have now revised our statistical analysis such that these constraints are not only implicit in our thinking and discussion, but explicit in the analysis itself: we have now combined linear mixed effects models with a step down approach that protects us against false positives.

1) In step 1, we perform only exactly 10 statistical comparisons as listed above using 2 (shape parameter: constant, linear) x 2 (ROI: primary, non-primary) LME models and 2 dimensional F-tests that assessed whether (i) the constant parameter (averaged across the 2 ROIs) or (ii) the linear parameter (averaged across the two ROIs) is significantly different from zero.

2) Only if this F-test is significant, we test separately for the linear and the constant parameters (each averaged across the two ROIs) whether they are significantly different from zero.

3) Only if this F-test is significant, we test separately for each ROI, whether the linear (or constant) is significantly different from zero in separate t-tests.

For the crossmodal deactivations, we use directed contrasts in step 2 and 3 to accommodate our apriori hypothesis that non-preferred stimuli induce deactivations.

This is described in a new Materials and methods section: Statistical analysis for the shape parameters at the between subject i.e. group level.

Tables 1, 2, 3 and Supplementary files 3 and 4 have been updated to reflect this change. All p-values that were effectively computed are now only displayed in the tables.

2) None of the effects in the main conclusion are reproduced in control task conditions: The attention effect in visual areas is not reproduced by attention effects in auditory areas. The cross-modal modulation in auditory areas is not reproduced in visual areas (neither in size, direction, nor shape of layer signals).

Thanks for this comment. Neurophysiological studies have shown that auditory influences on primary visual cortices rely on different translaminar mechanisms than visual influences on auditory cortices. Likewise, modality-specific attention is known to affect auditory and visual processing differently. Therefore, we would expect different laminar profiles in visual and auditory cortices.

As a result, visual areas cannot be considered control regions for auditory areas and vice versa. Instead, we would like to highlight that A1 and PT show comparable profiles for the visual induced deactivations and crossmodal modulations as the key effects reported in our study. In fact, laminar profiles for all effects are comparable for A1 and PT as well as V1 and V2/3 and can be considered as within study replication.

3) Similarly, none of the effects in the main conclusion are reproduced across the alternative control analysis approaches (shape parameters vs. decodability): Crossmodal layer-profiles in A1 and V1 (Figure 3A) look very different for B-parameters compared to the respective decodability values (zero, constant, decreasing, zero).

Thanks. Laminar BOLD profile and laminar decoding profile should not be viewed as control for one another. They focus on BOLD-response profiles that are expressed at different spatial scales and can therefore provide different results. For instance, if half of the ROI is activated and the other half is deactivated, the average ROI response will be zero, but MVPA will be able to discriminate between these two activation patterns. Likewise, one can simulate activation patterns that differ in their mean BOLD-response, but cannot be discriminated with MVPA. For instance, in our analysis we mean centre each feature individually before MVPA, which can easily lead to this pattern. Hence, ROI mean and ROI activation pattern analyses are complementary approaches to characterize our data and not expected to show comparable results.

These and other points are elaborated in the individual reviews below.Reviewer #1:The authors present an elegant study of visual, auditory and multimodal processing of looming stimuli. In addition, they employ an attention modulation, effectively generating a 3x2 design. They record high resolution (0.75mm isotropic) gradient-echo BOLD of 11 subjects and analyze it using the GLM framework. They use equivolume layering and the median activation over four ROIs (two auditory and two visual) to extract 4x6 laminar profiles. Summarizing their findings, they generally find the expected increase in visual areas for visual and auditory areas for auditory stimuli. They in addition find deactivations across modalities. They find cross-modal modulations only in auditory cortex, interestingly in Heschel's gyrus only in multi-voxel decoding, not in amplitude. Attentional modulations were mostly constrained to auditory regions as well.The study is complex, but with a solid design. The analyses are described in detail, the code is freely available (the data not yet, so I didn't test their pipeline), and the methods used are appropriate. The paper is also well written, clear and consistent. The figures are detailed, give single subject estimates and are mostly clear. I especially enjoyed that the authors provide all results, not only the significant tests. This is even more important, given that they test quite a lot (which should probably be discussed a bit more, especially in light of a small n=11), and thus there is no hard significance-filter for the results.

Thanks. This is an important point. Indeed, we included many tests. As explained in our response under ‘essential revision’ we implicitly imposed additional constraints to ensure that the key effects that we interpret are robust. Further, we would like to emphasize that we maximized within-subject reliability with a block design and acquiring 1200 functional volumes in total per subject. In our summary statistical approach, the variability (and hence statistics) at the 2^nd^ level implicitly combines within and between subject variability.

Reviewer #2:[…]– I feel the task design might have been pushed it a bit too much. There are as many as 6 task conditions with subtle differences. Thus, either one of the condition differentiations does not have so many trials to average across compared to comparable layer-fMRI studies. As a result, multiple different tasks conditions needed to be averaged together and the main conclusions are based on effects that (almost) disappear in the noise level.

While we appreciate the reviewer’s concerns, we have relied on a standard 2 x 3 factorial design. The strength of a factorial design is that in the absence of significant interactions we can pool over one factor (e.g. attention) to assess effects of the other factor making factorial designs flexible and powerful.

– I believe the clarity of the manuscript can be improved. Each figure has up to 20 sub panels, whereas most of them show insignificant effects that cannot be used to support the main conclusion. It took me quite a while to filter out the relevant information.

We have now improved our figures and highlighted the key effects we want to focus on. We hope that this improves the overall readability of the results.

Figure 2 has been split into two figures. Because we are not interested in the ‘obvious’ activations for the preferred sensory modality we have moved those results to a separate supplementary figure. Similarly Figure 3 has been split into two figures and the results from the visual areas have been put into a supplementary figure. Figure 4 has been reorganized to hopefully be made more readable.

– I believe the way how the analysis is conducted and the data are presented could benefit from more discussions on the limits of their interpretability. I am hesitant whether I can interpret the shape parameters and decodability profiles as measures of neural activity in a way that the main conclusion is the most plausible explanation for all the depicted results.

Thanks. We have now included a critical discussion of our approach.

Subsection “Shape parameters for characterization of laminar BOLD response profiles”: “The S-parameters enable us to make inferences about the shape of a laminar BOLD-response profile rather than using an omnibus F-test that assesses whether the BOLD-response differs across any of the six laminae followed by numerous post hoc pairwise comparisons between layers. However, we acknowledge that this characterization makes our inference less specific about locating even the BOLD-response to a particular lamina. Moreover, because the laminar GLM in the current report did not include any higher order quadratic etc. terms they would not be to model U-shaped laminar profiles.”

Subsection “Multivariate analysis of pattern across vertices and decoding accuracy profiles”: “Please note that the laminar profiles of decoding accuracy need to be interpreted cautiously. First, decoding accuracy depends on BOLD-signal and noise characteristics that can both vary across laminae. Second, the laminar profile of decoding accuracy is more difficult to interpret, because accuracy is bounded between 0 and 1.”

Reviewer #3:In this paper, the authors examine the responses of early sensory cortices to auditory, visual, and combined auditory-visual stimuli, in conjunction with an attentional modulation, using laminar resolution fMRI. The results presented are of interest, and convincing. Nevertheless, I feel that the paper could be improved by consideration of the following:1) In the Abstract the claim is made that these findings are crucial for understanding "how the brain regulates information flow across senses". A sceptic could claim that the article only succeeds in reproducing animal literature, and I would suggest that the authors expand their discussion so as to better substantiate this claim.

Thanks. We have now expanded the Discussion to compare with previous rodent literature. However, given the enormous differences between rodents and humans, we feel even a simple replication would be sufficient to excite the reader.

Subsection “Multisensory effects in visual and auditory cortices – a comparison”: “Our results reveal distinct laminar profiles for multisensory deactivations in auditory and visual cortices. […]Finally, interpreting laminar BOLD-response profiles in relation to previous findings in rodents remains tentative because of cross-species and methodological differences.”

2) The figures are organised to present the results of auditory and visual stimulation separately, whereas the text is organised to first deal with auditory and then visual stimulation. It may be more logical to deal with both modalities in parallel. Beyond the criticism at the organisational level, the current structure rather masks obvious asymmetries between the responses in primary auditory and visual cortices. The narrative that emerges would seem to be of a dominance visual modality. The Discussion rather misses the opportunity to compare and contrast across the modalities. For example, there are obvious differences shown in Figures 2A, and 3A which could benefit from some in-depth discussion.

Thanks for this suggestion. In fact, we had initially explored an organization based on effect rather than anatomical region. However, this direct opposition seemed to make the Discussion a little cumbersome. Moreover, we are cautious about directly comparing the effects we found in auditory cortices with those not found in visual cortices, for two reasons: First, we cannot exclude the possibility that we observe effects in auditory cortices, but not in visual cortices because of differences in vascular organization. Second, we are cautious about inferring directly a dominance of one sensory modality, because we did not elaborately fine tune the stimuli. For instance, in the past it was thought that vision dominated audition in spatial perception. But research has now established that in fact this is just a question of signal reliability. If we degraded visual information, potentially we may observe that audition dominates vision. Hence, we feel it is premature to draw strong inferences about sensory dominance.

We have now included a brief paragraph comparing effects in visual and auditory cortices in the Discussion, please see above.

3) Cross modal modulation is assessed by examining the contrast AV-A in auditory cortex and correspondingly AV-V in visual cortex. To me it would seem more logical to construct a contrast AV-(A + V). For example, if AV-A would produce no significant activation, but a visual stimulus in isolation would deactivate auditory cortex, then there would be a clear cross-modal effect that would be missed by your chosen contrast parameter. I would suggest re-analysing the data.

The reviewer is absolutely correct that many studies investigating audiovisual integration report audiovisual interactions, i.e. AV-(A + V). However, as we have discussed in greater detail in Noppeney, 2012, audiovisual interactions in particular when observers are engaged in a task are difficult to interpret. For instance, attentional and response selection processes will be counted once for AV, but twice for the sum A+V. A related issue occurred to us when we tried to perform the decoding analysis of [AV vs. A+V] where we realized that the results of such classifications were mostly driven by differences in noise, because the variance of the sum A+V differ from AV even under the null-hypothesis of no difference. As a result, discriminating between AV vs. A+V would not be valid with MVPA. In this study, we have therefore taken a different approach. First, we focus on crossmodal (de)activations in a unisensory context. Second, we investigate how a concurrent visual signal modulates the response to an auditory signal in auditory cortices (and vice versa in visual cortices). The latter contrast is also a standard statistical comparison in the multisensory integration community. The basic rationale of both statistical comparisons is: If primary and secondary auditory (resp. visual) cortices are specific to processing auditory (resp. visual) stimuli we should not observe and difference.

4) Please justify the inclusion of, and discuss the interpretation of the results from, the planum temporale. In this context please also explain in the Results why the left hemisphere was imaged.

We are not sure whether we understand the reviewer correctly. Both hemispheres were imaged; we presented the laminar profiles results pooled over both hemispheres. A sample of individual surface projections from both hemispheres is now presented in the Figure 2—figure supplement 2 and 3 to clarify this point.

The passages below illustrate where some of the instances where this fact is now mentioned:

– The MRI data acquisition section:

“The acquisition slab was inclined and positioned to include Heschl’s gyri, the posterior portions of superior temporal gyri and the calcarine sulci. Both hemispheres were acquired. On the second day the slab was auto-aligned on the acquisition slab of the first day. This provided us with nearly full coverage over all our four regions of interest.”

– The sampling of the BOLD-response along cortical depth section:

“The activity values at the vertices of these 6 intra-cortical surfaces in our four ROIs (i.e. primary auditory, planum temporale, V1, V2/3) pooled over both hemispheres form the basis for our univariate ROI analysis of the laminar BOLD-response profiles and laminar decoding accuracy profiles (based on multivariate pattern classification).”

5) It could be easier to follow some of the text if the authors would deal with areas of early visual cortex that activate upon visual stimulus separately from those that deactivate upon visual stimulus.

One of our initial analysis did investigate the shape of laminar profile in response to visual stimuli in the sub-regions activated VS deactivated by visual stimuli but we found that this could lead to biased laminar profile shapes due to double dipping issues. We have therefore decided not to present such results.

6) I had some concerns about the stimulus design, which I could not find addressed in the text. First, the baseline condition of visual fixation is itself some form of visual task.

Participants fixated a central cross in all conditions, i.e. A, V, AV and Baseline. Because the baseline is not modelled, we subtract all baseline processes implicitly from all conditions. The following sentence was added to the Experimental procedures section:

“Throughout the entire experiment, participants fixated a cross (0.16º visual angle) in the centre of the black screen. Therefore, processes related to fixation are present in all conditions (i.e. all stimulation and fixation conditions). “

Second, the auditory target apparently has a fixed amplitude but is presented at a variable time against a looming auditory stimulus: doesn't this affect the detectability?

This is correct. We have made this choice to equate the visual and auditory tasks:

– Visual targets of a fixed radius could also be presented at increasing at levels of visual eccentricities during the duration of the visual stimulus (“The visual grey dot could be presented at any time at any location within the area defined by the radially expanding outermost annulus until they reach a radius of maximal 3.88º visual angle.”) hence visual targets presented later during a visual stimulus would also be harder to detect.

– Auditory target intensity and visual target radius were adapted for each subject to try to equate their detectability across conditions (“The size of the visual target and the sound amplitude of the auditory target were adjusted in a subject-specific fashion to match the approximate target detection performance across sensory modalities and then held constant across the entire experiment. This was done using the method of constant stimuli in a brief ‘psychometric function experiment’, performed prior to the fMRI study inside the scanner and with scanner noise present.”)

7) The English is generally good, but the authors persistently introduce comparative statements with no clear object. This is particularly confusing when paragraphs start with: "by contrast"; "in contrast"; "hence". Most of these can be deleted without any effect on the meaning. The authors also tend to introduce a chronology ("next we…") which is not really necessary.

Thanks. Following the reviewer’s advice, we have removed all these words.